# Complement-mediated enhancement of SARS-CoV-2 antibody neutralisation potency in vaccinated individuals

Jack Mellors [1] ✉, Raman Dhaliwal[2], Stephanie Longet[3], Tom Tipton[1], OCTAVE Consortium*, OPTIC Consortium*, Eleanor Barnes[4,5], Susanna J. Dunachie[4,6], Paul Klenerman[4,5], Julian Hiscox[7] & Miles Carroll [1] ✉

With the continued emergence of SARS-CoV-2 variants and concerns of waning immunity, there is a need for better defined correlates of protection to aid future vaccine and therapeutic developments. Whilst neutralising antibody titres are associated with protection, these are typically determined in the absence of the complement system, which has the potential to enhance neutralisation titres and strengthen correlates with protection in vivo. Here we show that replenishment of the complement system in neutralisation assays can significantly enhance neutralisation titres, with up to an ~83-fold increase in neutralisation of the BA.1.1.529 strain using cross-reactive sera from vaccination against the ancestral strain. The magnitude of enhancement significantly varies between individuals, viral strains (wild-type/VIC01 and Omicron/BA.1), and cell lines (Vero E6 and Calu-3), and is abrogated following heat-inactivation of the complement source. Utilising ACE2 competition assays, we show that the mechanism of action is partially mediated by reducing ACE2-spike interactions. Through the addition of compstatin (a C3 inhibitor) to live virus neutralisation assays, the complement protein C3 is shown to be required for maximum efficiency. These findings further our understanding of SARS-CoV-2 immunity and neutralisation, with implications for protection against emerging variants and assessing future vaccine and therapeutic developments.

The implementation of COVID-19 vaccines has proven highly effective against the development of severe disease, hospitalisation, and death. There is a good correlation between antibody binding and antibody neutralisation with protection against disease, but this can change within the context of viral evolution and emerging variants[1], with further complexity in correlating the impact of Fc effector functions[2]. With concerns regarding breakthrough infections, a lack of therapeutics, and ongoing attempts to develop vaccines to combat the continued emergence of new variants, clearly defined and ongoing assessments of correlates of protection are imperative.

[1]Centre for Human Genetics and the Pandemic Sciences Institute, Nuffield Department of Medicine, University of Oxford, Oxford, UK. [2]Sir William Dunn School of Pathology, University of Oxford, Oxford, UK. [3]Centre International de Recherche en Infectiologie, Université Jean Monnet, Université Claude Bernard Lyon, Inserm, Saint-Etienne, France. [4]NIHR Oxford Biomedical Research Centre, Oxford University Hospitals NHS Foundation Trust, Oxford, UK. [5]Translational Gastroenterology and Liver Unit, Nuffield Department of Medicine, University of Oxford, Oxford, UK. [6]NDM Centre for Global Health Research, Nuffield Department of Medicine, University of Oxford, Oxford, UK. [7]Department of Infection Biology and Microbiomes, Institute of Infection, Veterinary and Ecological Sciences, University of Liverpool, Liverpool, UK. *Lists of authors and their affiliations appear at the end of the paper. ✉e-mail: jmellors@ic.ac.uk; miles.carroll@ndm.ox.ac.uk

The complement system – comprised of heat-labile plasma proteins which form part of the innate immune response – can enhance the potency of neutralising antibodies in vitro and strengthen the relationship of neutralisation titres with protection in vivo[3–5]. Some antibodies are entirely dependent on the complement system for virus neutralisation[6–11]. A complement-mediated enhancement of neutralising antibody titres has been shown against a range of viruses including cytomegalovirus[9,11–18], Ebola virus[19], influenza virus[4,5,20,21], and vaccinia virus[22–25]. This phenomenon is independent of other immune functions such as opsonisation and phagocytosis[26]. Despite its significance, the complement system in sera/plasma is typically inactivated, or is poorly conserved, prior to its use in neutralisation assays. This can be due to common practices such as the heat inactivation of samples at ≥56 °C, or the use of anticoagulants such as ethylenediaminetetraacetic acid (EDTA) during blood collection. Despite the widespread use of conventional neutralisation assays for SARS-CoV-2 immunity research, a complement-mediated enhancement of SARS-CoV-2 neutralising antibody titres has not yet been reported.

Activation of the complement system can occur via three distinct pathways: classical, lectin, and alternative. The classical pathway is typically activated via the binding of the C1 protein (C1q protein in complex with C1r and C1s proteases) to antibodies in complex with the viral antigen. This causes the proteolytic cleavage of C4 and C2 to form the C3 convertase (C4b2a). The C3 convertase then cleaves C3 into C3a (anaphylatoxin) and C3b to form the C5 convertase (C4b2a3b). C5 is then cleaved into C5a (anaphylatoxin) and C5b which enables subsequent binding of C6, C7, C8, and multiple copies of C9 to form the membrane attack complex (MAC). The lectin pathway differs in its activation, with the binding of pattern recognition molecules (PRMs) such as mannose binding lectin (MBL) to glycosylated regions of the viral antigens. MBL-associated serine proteases (MASPs) in complex with the PRMs then mediate cleavage of C4 and C2, before following the same protein cascade as the classical pathway. Lastly, the alternative pathway is typically activated via the spontaneous hydrolysis of C3. The remaining C3b molecule, in the absence of complement regulatory proteins, binds to factor B and is subsequently cleaved by factor D to form the C3bBb complex. The binding of properdin then stabilises the C3bBb complex which is capable of cleaving C5. Activation of the alternative pathway can therefore augment the classical and lectin pathways or function independently[26,27].

There are four commonly reported mechanisms which explain the complement-mediated enhancement of neutralising antibody potency, independent of other immune functions such as opsonisation[26]. As the process is antibody-mediated, these mechanisms pertain to the classical pathway and include: the aggregation of virus particles, the inhibition of viral attachment/entry to host-cell receptors, the lysis of virus particles, or the lysis of infected cells. The aggregation of virus particles by antibody binding can cause a reduction in viral attachment to host cells. The formation of viral aggregates can be enhanced by the deposition of complement proteins following antibody binding, which usually depends on proteins C1–C3[5,8,24]. The second mechanism, the inhibition of viral attachment/entry to host cells, refers to the masking of the viral antigens required for infection through the deposition of complement proteins. In addition to antibody binding, the subsequent binding of C1, C4, C2, and multiple C3 molecules (where up to 1000 C3 molecules may be cleaved by one C3 convertase) increases the chances of blocking protein–protein interactions required for infection[2,4,28,29] or reducing the stoichiometric threshold for antibody-mediated neutralisation[30]. The third mechanism of viral lysis requires the complete activation of the complement system, resulting in the formation of the MAC. The MAC may lyse the lipid membranes of enveloped viruses, thus reducing their infectivity[31,32]. The fourth mechanism also depends on complete activation of the complement system. Antibodies can bind to viral antigens expressed on the surface of infected cells, leading to complement deposition and formation of the MAC to lyse the infected host cells and reduce viral titres[21,24].

To determine whether the complement system can enhance the SARS-CoV-2 neutralising antibody titres of COVID-19 vaccine recipients, in this work we use microneutralisation assays (MNAs) with sera from two vaccine cohorts and supplement them with exogenous pooled human plasma (PHP) as a complement source. We find that the presence of the complement system significantly enhances SARS-CoV-2 neutralisation titres against wild-type virus (VIC01 strain) by up to 20-fold and enhances cross-reactive neutralisation of the Omicron BA.1 strain by up to 83-fold. In some instances, neutralisation of the BA.1 strain is entirely complement-dependent. The magnitude of these responses differs depending on the viral strain (VIC01 or BA.1), the cell line (Vero E6 or Calu-3), and the individual immune sera. This response is significantly diminished following heat inactivation of the complement source with a loss in neutralisation of up to 59%, it is partially mediated by the inhibition of ACE2-spike interactions, and the complement protein C3 is required for maximum efficiency.

## Results

### The addition of PHP significantly increased the SARS-CoV-2 neutralising antibody titres of the OPTIC cohort

The addition of PHP to the MNAs significantly increased the 50% neutralisation titres (NT50s) of the OPTIC cohort vaccinee serum samples compared to the addition of HI-FCS or media-only. The enhancement in NT50 was observed for all conditions: Vero E6 cells infected with VIC01 (Fig. 1a); Vero E6 cells infected with BA.1 (Fig. 1b); Calu-3 cells infected with VIC01 (Fig. 1c); and Calu-3 cells infected with BA.1 (Fig. 1d). There was no significant difference between NT50s with the addition of media-only or HI-FCS controls. Also, the use of HI-FCS or PHP did not demonstrate virus neutralising activity in the absence of OPTIC vaccinee serum (Supplementary Fig. 3) nor cytotoxicity (Supplementary Fig. 4). These results show an antibody-mediated effect that is enhanced only with the use of non-heat-inactivated PHP, which is indicative of complement activity. In some instances, as demonstrated in Fig. 1d with the infection of Calu-3 cells using the BA.1 strain, the presence of PHP was essential for SARS-CoV-2 neutralisation to be detected. As the OPTIC serum samples were collected prior to the emergence of BA.1, these results show that the addition of PHP could enhance, and in some instances was essential for, the cross-neutralisation of BA.1.

The effect of PHP compared to HI-FCS was used as an indication of complement activity and represented as a log2 fold-change (log2FC) (Fig. 1e). The extent of complement utilisation varied between cell lines and/or viral strains. The addition of PHP significantly enhanced NT50s against both the VIC01 (average log2FC 0.96) and BA.1 (average log2FC 0.99) strains when using Vero E6 cells. There was no significant difference between the enhanced neutralisation of these two viral strains in this condition ($p = 0.9996$). Enhancement of NT50s was significantly higher in the Calu-3 cells compared to Vero E6 cells for both the VIC01 (average log2FC 3.04) and BA.1 (average log2FC 4.14) strains. Furthermore, there was significantly greater enhancement of NT50s against the BA.1 strain compared to the VIC01 strain when using Calu-3 cells (average log2FC of 1.10). Cell line differences for NT50s were also observed in the media-only controls, with lower average NT50s for calu-3 cells versus Vero E6 cells (Fig. 1a–d and Supplementary Data 1). All NT50 values for the OPTIC cohort can be found in Supplementary Data 1.

Lastly, the enhancement in NT50s against the BA.1 strain with the addition of PHP only occurred in some serum samples within the cohort, which again varied between cell lines (Fig. 1f). This shows a serum-specific response to the utilisation of complement, which is further described in the next section. Overall, these results show that the presence of complement enhances NT50s against the VIC01 and BA.1 strains in a manner that is antibody dependent and varies in magnitude

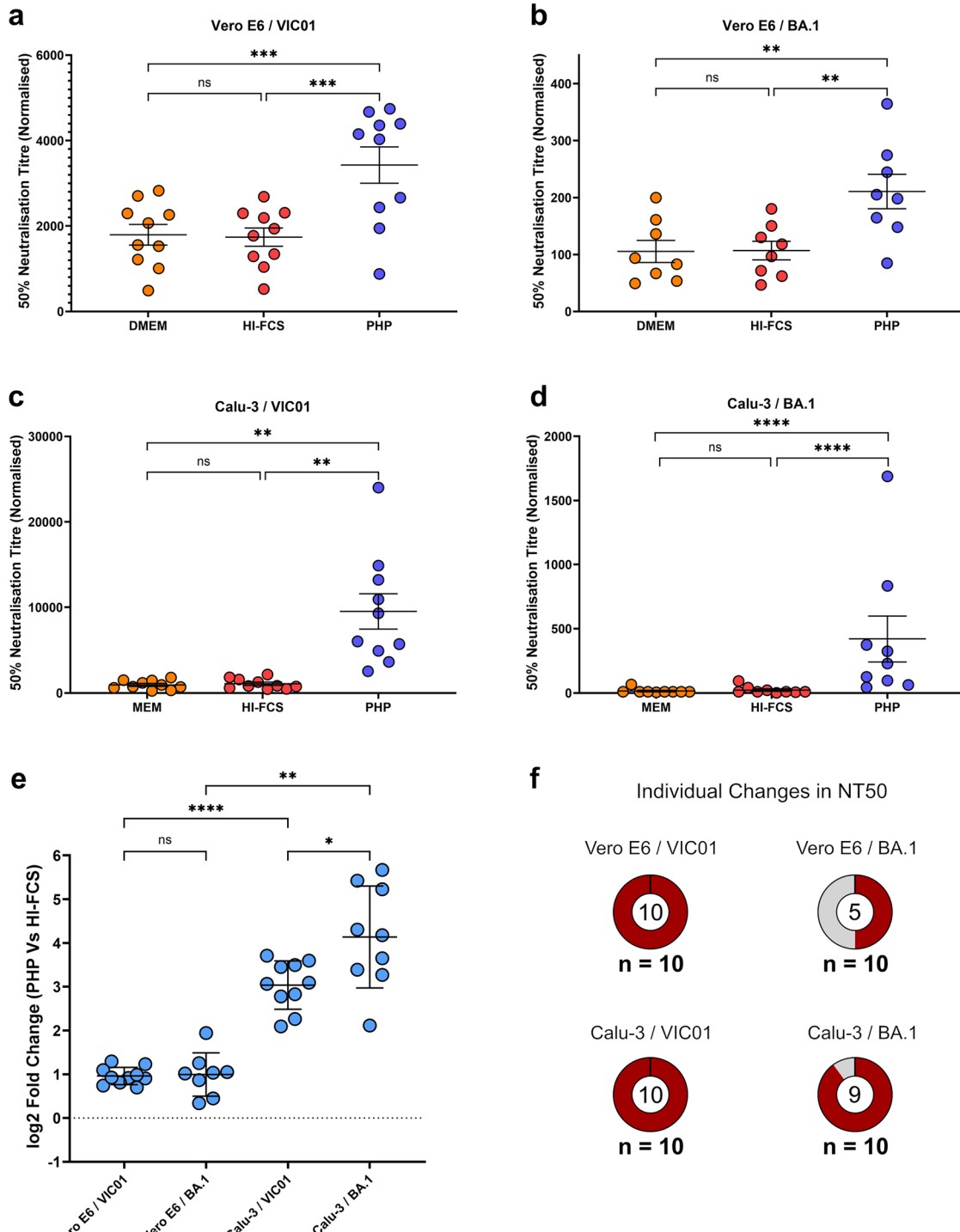

depending on the immune sera and cell lines used. These variations suggest the importance of antibody characteristics, viral epitopes, and host-cell receptors for virus entry in determining this response.

### Enhancement with PHP was heat labile and occurred during, or prior to, the early stages of infection

PHP was tested in parallel both pre- and post-heat inactivation, to determine whether the enhancement of NT50s with the OPTIC vaccinee serum samples ($n = 10$) was heat labile. All OPTIC vaccinee serum samples diluted to 1:1500 (Fig. 2a) and 1:4500 (Fig. 2b) showed a significant loss in neutralisation following the heat inactivation of PHP (excluding sample 9 at the 1:4500 dilution which was close to the limit

of detection), with a change in neutralisation of up to 59%. This shows that the enhancement of antibody-mediated neutralisation was heat labile, as expected for complement activity. Furthermore, the addition of PHP improved the assay sensitivity for some samples, which would otherwise demonstrate no neutralising activity at the 1:4500 dilution (Fig. 2b).

The supernatant (containing virus particles, OPTIC vaccinee serum, and PHP/HI-PHP) was incubated for 1 h in the absence of cells, followed by a 2-h incubation step in the presence of Calu-3 cells, before the supernatant was removed and replaced with fresh media. Therefore, the enhancement of antibody-mediated neutralisation occurred prior to, or within the early stages of, Calu-3 cell infection.

**Fig. 1 | The presence of the complement system significantly increased SARS-CoV-2 neutralisation titres in the OPTIC vaccinee cohort.** 50% neutralisation titres (NT50) were determined via microneutralisation assays for all OPTIC vaccinee serum samples following the addition of media-only (DMEM/MEM), heat-inactivated (HI)-FCS, or pooled human plasma (PHP). **a** Vero E6 cells infected with VIC01 ($n = 10$) (PHP vs. DMEM, $p = 0.0002$; PHP vs. HI-FCS, $p = 0.0002$); **b** Vero E6 cells infected with BA.1 ($n = 8$) (PHP vs. DMEM, $p = 0.0070$; PHP vs. HI-FCS, $p = 0.0063$); **c** Calu-3 cells infected with VIC01 ($n = 10$) (PHP vs. MEM, $p = 0.0050$; PHP vs. HI-FCS, $p = 0.0043$); **d** Calu-3 cells infected with BA.1 ($n = 9$) (PHP vs. DMEM, $p = <0.0001$; PHP vs. HI-FCS, $p = <0.0001$). Each spot shows the average NT50 value for each sample determined by a 4-parameter logistic curve from four replicates across duplicate assays. Error bars show the mean with the standard error. Significance was determined using a one-way ANOVA with Geisser-Greenhouse correction and Tukey's multiple comparisons test. **e** Log2 fold-changes in NT50 between HI-FCS and PHP represents the enhancement of neutralisation via the complement system for each condition shown in (**a**–**d**) (Vero E6/BA.1 vs. Calu-3/BA.1, $p = 0.0015$; Calu-3/VIC01 Vs Calu-3/BA.1, $p = 0.0210$; Calu-3/VIC01 Vs. Vero E6/VIC01, $p = <0.0001$). Statistical significance was determined using a one-way ANOVA mixed effects analysis with Geisser-Greenhouse correction and Šídák's multiple comparisons test. Error bars show the mean value with standard deviation. For (**a**–**e**), arbitrary values of 10 were used for samples with a predicted NT50 below this value. If an NT50 value could not be determined in any condition, then the sample was omitted. **f** Significant differences in NT50 were determined for each individual using the sum-of-squares $F$-test with non-overlapping 95% confidence intervals. The outside number shows the total sample size and the centre number (shown as a percentage in red) reports the number of individuals with a significant increase in NT50 following the addition of PHP. All results were analysed and presented using GraphPad Prism (Version 10) and Inkscape. $*p < 0.05$, $**p < 0.01$, $***p < 0.001$, $****p < 0.0001$. Source data are provided as a Source Data file.

## Enhancement was likely the result of blocking virus interactions with host-cell receptors

Four mechanisms for the complement-mediated enhancement of neutralising antibody titres are predominantly discussed in the literature. They are: the aggregation of virus particles, the blocking of virus-host receptor interactions, and the MAC-mediated lysis of virus particles or infected cells. The cell line differences observed previously would suggest that the blocking of virus-host receptor interactions is the most plausible explanation for the observations within this study.

Transmission electron microscopy was used to determine whether viral lysis and/or the aggregation of virus particles may be responsible for the complement-mediated enhancement of neutralisation titres. Negative staining of the samples revealed that the addition of PHP did not noticeably differ in the lysis of virus particles or the formation of viral aggregates from the use of HI-PHP, nor the virus-only control, using OPTIC serum samples 8 (Fig. 3a) and 10 (Supplementary Fig. 5). Based on the four proposed mechanisms, the blocking of virus-host receptor interactions would be consistent with these findings.

Compstatin is a selective inhibitor of the complement pathway which binds the C3 protein and prevents the proteolytic cleavage required for its activation. The compstatin control peptide is a negative control for compstatin. The use of compstatin or the compstatin control peptide resulted in significant increases in neutralisation titres compared to the immune sera alone (Fig. 3b). However, in 2/3 of the samples tested, the effects of compstatin on neutralisation were significantly lower than the control peptide. The third sample was not significant but shows the same trend. These results show that C3 is required for the full efficiency of the complement system to promote neutralisation, but it is not essential to still see enhancement. These results further support the proposed mechanism of blocking virus-host receptor interactions, which is typically achieved using proteins C1–C3, and suggests that partial enhancement of neutralisation may be obtained using a combination of proteins C1, C4, and C2. Whilst we cannot definitively say that all C3 proteins within the samples were inhibited, compstatin and the control peptide were administered at double the reported IC50 values for physiological concentrations, and complement was used at 20% of this physiological concentration, so the inhibitor and control peptide were likely in excess.

ACE2 inhibition assays were then used to determine whether the addition of PHP could enhance the antibody-mediated inhibition of SARS-CoV-2 spike protein interactions with a recombinant human ACE2 protein. For all SARS-CoV-2 antigens tested, the use of PHP significantly increased the levels of ACE2 inhibition compared to the use of HI-PHP (Fig. 3c) with the OPTIC samples tested (samples 1, 2, 7, and 8). Although the addition of PHP to immune sera enhanced ACE2 inhibition, the extent of enhancement was much lower than what was observed for the neutralisation assays. This suggests that the complement-mediated enhancement of ACE2-spike inhibition may be a partial mechanistic explanation for the enhancement of neutralisation.

## Fold-enhancement of neutralising antibody titres using PHP varied between sample cohorts

The OCTAVE cohort (vaccinated, immunocompromised individuals) provided further comparison of the effects of PHP between immune serum samples. A significant increase ($p = 0.0003$) in SARS-CoV-2 VIC01 neutralisation with the addition of PHP compared to HI-FCS, was observed for the OCTAVE vaccinee cohort using Vero E6 cells (Fig. 4a). However, only four of the twenty-one samples (19%) within this cohort showed a significant increase in NT50 across three independent experiments measured by the sum-of-squares $F$-test with non-overlapping 95% CIs. In comparison, all ten samples in the OPTIC cohort (100%) showed a significant increase with the addition of PHP under the same conditions (Fig. 1f). Again, a log2 fold increase was used to show the effects of PHP compared to HI-FCS as an indication of complement activity, and the enhancement of NT50s were significantly greater in the OPTIC cohort (Fig. 4b). All NT50 values for the OCTAVE cohort can be found in Supplementary Data 2. These results further suggest that the complement-mediated enhancement is not only antibody dependent, but it is specific to certain immune serum samples. This could be due to the epitope specificity of the antibodies, antibody glycosylation, IgG subclass, and antibody isotype. It is unclear which differences between the OPTIC and OCTAVE cohorts may be responsible for this, as the cohorts were not matched on factors including health status, vaccine status, time of sample collection, age, or sex. Of the four OCTAVE samples showing a significant complement-mediated change in NT50, three were diagnosed with liver cirrhosis (33% of total cohort, $n = 21$) and one was diagnosed with ulcerative colitis (17% of total cohort, $n = 21$).

## Comparisons of antibody characteristics associated with a complement-mediated enhancement of neutralisation

There are various antibody characteristics which could explain the observed differences between donors, including: epitope specificity, glycosylation, affinity, isotype, and IgG subclass[26]. Samples from both the OPTIC and OCTAVE cohorts were categorised depending on whether a significant enhancement in neutralisation against VIC01 using Vero E6 cells occurred (Enhanced Cohort, $n = 13$) or not (Non-Enhanced Cohort, $n = 17$).

IgG1 and IgG3 subclasses are reportedly the most potent activators of the complement system, followed by IgG2, then IgG4 with minimal activity reported. Firstly, total anti-SARS-CoV-2 spike IgG titres were significantly higher in the Enhanced cohort ($p = 0.0029$) (Fig. 5a). Both cohorts showed the same trend with the highest measurements for IgG1 binding, followed by IgG2, then IgG3, then IgG4 (Fig. 5b), with significantly higher IgG1 ($p = 0.0329$), IgG2 ($p = 0.0098$), and IgG3 ($p = <0.0001$) titres in the Enhanced group (Fig. 5c). In spite of the

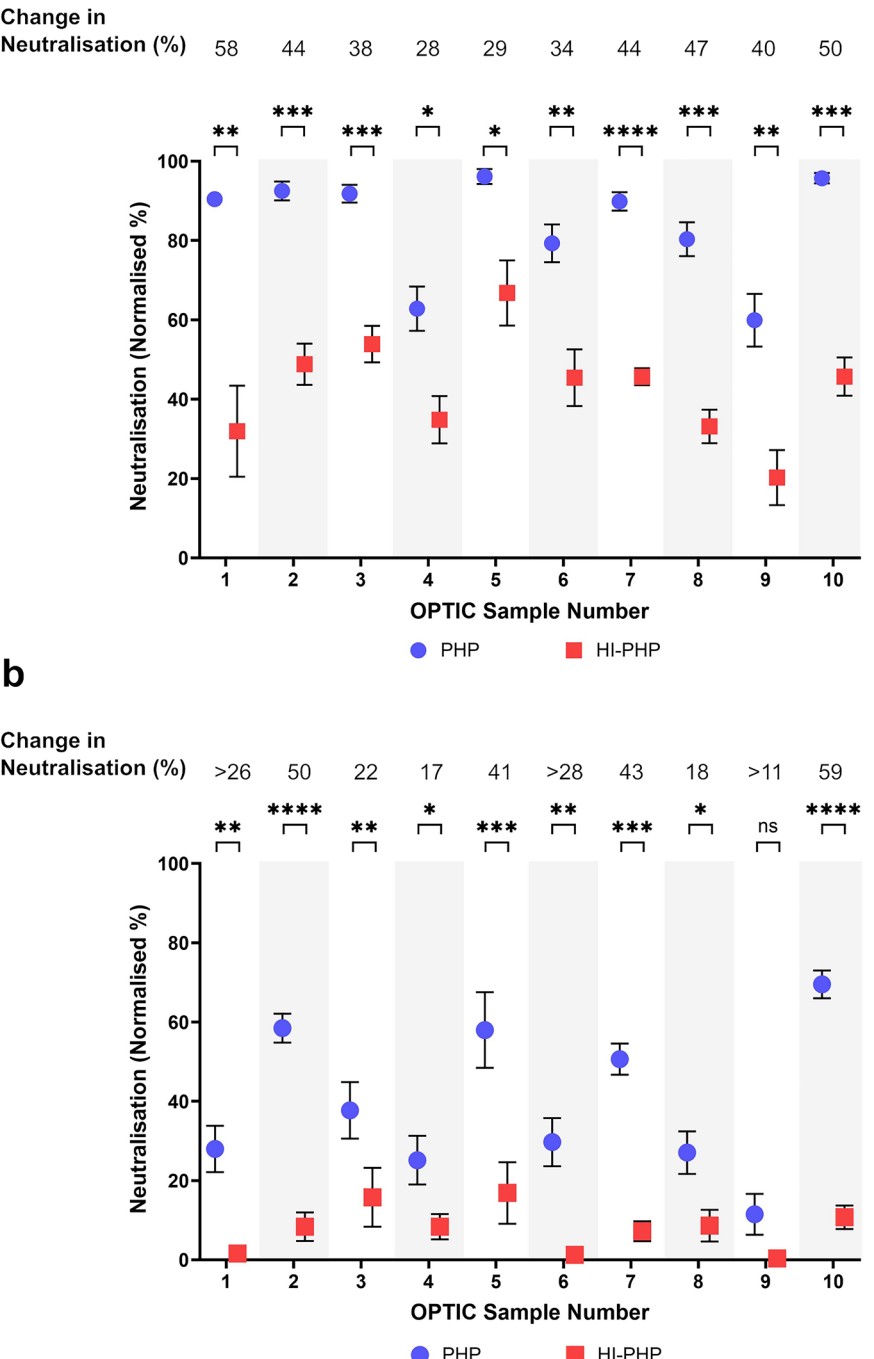

**Fig. 2 | The enhancement of SARS-CoV-2 neutralisation titres in the presence of the complement system was heat labile.** Comparison of SARS-CoV-2 (VIC01) neutralisation using 20% pooled human plasma (PHP) or 20% heat-inactivated (HI)-PHP for all vaccinee serum samples in the OPTIC cohort (*n* = 10) at a **a** 1:1500 and **b** 1:4500 dilution. Each dot is the average of six replicates across duplicate assays and the error bars show the standard error. All samples showed a significant decrease in SARS-CoV-2 neutralisation following the heat inactivation of PHP (paired, two-sided *T*-test, *p* < 0.05), excluding one sample with values close to the limit of detection. The results were analysed and presented using GraphPad Prism (Version 10). *\*p* < 0.05, *\*\*p* < 0.01, *\*\*\*p* < 0.001, *\*\*\*\*p* < 0.0001. Exact *p* values for **a** 1 (*p* = 0.0021), 2 (*p* = 0.0002), 3 (*p* = 0.0003), 4 (*p* = 0.0305), 5 (*p* = 0.0113), 6 (*p* = 0.0024), 7 (*p* = 0.0001), 8 (*p* = 0.0007), 9 (*p* = 0.0020), 10 (*p* = 0.0002) and **b** 1 (*p* = 0.0034), 2 (*p* = <0.0001), 3 (*p* = 0.0083), 4 (*p* = 0.0376), 5 (*p* = 0.0002), 6 (*p* = 0.0070), 7 (*p* = 0.0005), 8 (*p* = 0.0236), 9 (*p* = 0.0725), 10 (*p* = 0.0001). Source data are provided as a Source Data file.

differences in total IgG titres, there was no significant difference in IgG4 (*p* = 0.1875) titres, which suggests that this subclass constitutes a higher proportion of total IgG in the Non-Enhanced group. Lastly, antibody-dependent complement deposition (ADCD), which measured C3c deposition in response to the SARS-CoV-2 spike protein, was significantly higher in the Enhanced cohort (*p* = 0.0031) (Fig. 5d). To understand the relationships between total IgG, IgG subclass, ADCD, and complement-enhanced neutralisation, all conditions were correlated using Pearson correlations (Benjamini–Hochberg false discovery rate of 0.05) (Fig. 5e). The most significant correlations for the Non-

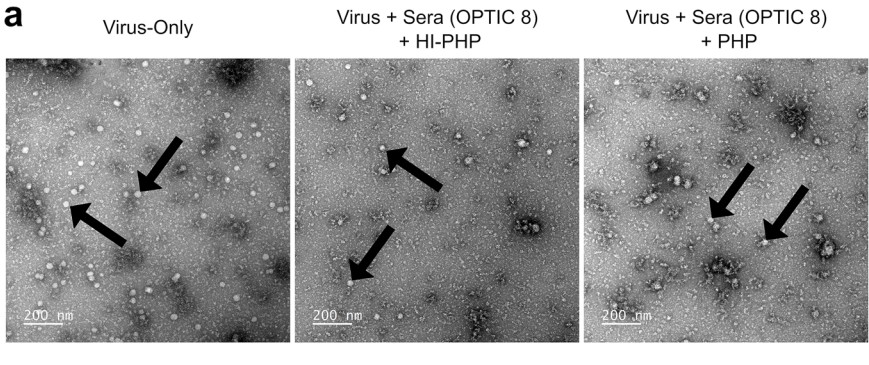

**a**

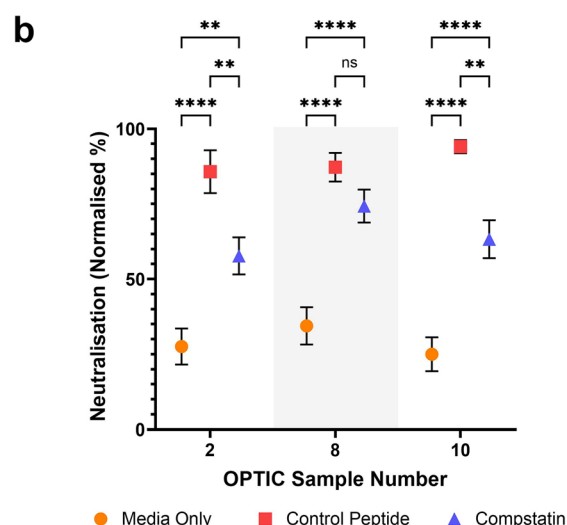

**b**

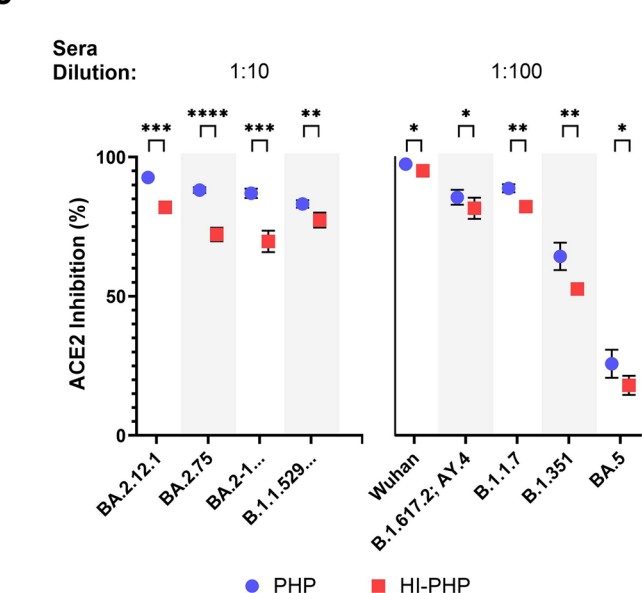

**c**

Enhanced cohort were for total IgG, ADCD, and neutralisation, whereas the Enhanced cohort was total IgG and neutralisation only. The Non-Enhanced cohort also showed significant correlations of IgG4 titres with neutralisation, which was not observed in the Enhanced cohort. There were no clear correlations that could distinguish between HI-FCS and PHP supplemented neutralisation titres in either cohort to provide insight into complement-mediated enhancement.

We also examined antibody isotype and/or epitope specificity to various Coronavirus antigens for the Enhanced and Non-Enhanced cohorts, using previously published data[33,34]. The most notable differences between the two cohorts were in the relationships of IgG titres specific to 229E and NL63 spike proteins against other Coronavirus antigens and with neutralisation, where IgG binding to the 229E-spike protein significantly correlated with neutralisation for the

**Fig. 3 | A complement-mediated mechanism of enhancement for SARS-CoV-2 neutralisation titres.** Use of transmission electron microscopy (TEM), compstatin, and ACE2 inhibition assays to determine the mechanism of complement-enhanced neutralisation. **a** TEM was used to identify possible viral aggregation and/or lysis following incubation with immune sera (using representative data from OPTIC sample 8) and pooled human plasma (PHP) or heat-inactivated (HI)-PHP. Each biological sample was tested in duplicate with a total of 136 images captured across three magnifications. No clear difference was observed between the conditions tested. The black arrows indicate examples of the SARS-CoV-2 (VIC01) particles. **b** Microneutralisation assays with compstatin or a control peptide showed the effects of C3 inhibition. Each spot represents the mean value of 6 replicates across duplicate assays and error bars show the standard error. The addition of PHP with either compstatin or the control peptide significantly increased SARS-CoV-2 neutralisation. A further increase in neutralisation was observed with the use of the control peptide, which was significant in 2/3 samples using a two-way ANOVA with Tukey's multiple comparisons test. Exact p values for samples 2 (media vs. control, p = <0.0001; media vs. compstatin, p = 0.0015; compstatin vs. control, p = 0.0032),

8 (media vs. control, p = <0.0001; media vs. compstatin, p = <0.0001; compstatin vs. control, p = 0.2562), and 10 (media vs. control, p = <0.0001; media vs. compstatin, p = <0.0001; compstatin vs. control, p = 0.0012) **c** Human ACE2 competition assays were supplemented with either PHP or HI-PHP to measure the effect of complement on ACE2 binding to various SARS-CoV-2 spike proteins. The presence of complement significantly enhanced ACE2 inhibition for all antigens tested. Each spot represents duplicate values of 4 OPTIC serum samples and the error bars show the standard error. Sample dilutions of either 1:10 or 1:100 are shown dependent on whether the observations were within the limits of detection. Significance was determined using paired, two-sided *T*-tests for each antigen. Exact p values are 0.0004 (BA.2.12.1), <0.0001 (BA.2.75), 0.0001 (BA.2-1...), 0.0041 (B.1.1.529), 0.0256 (Wuhan), 0.0302 (B.1.617.2;AY.4), 0.0035 (B.1.1.7), 0.0085 (B.1.351), 0.0334 (BA.5). The results were analysed and presented using GraphPad Prism (Version 10). *p < 0.05, **p < 0.01, ***p < 0.001, ****p < 0.0001. Antigen "BA.2-1..." includes: BA.2; BA.2.1; BA.2.2; BA.2.3; BA.2.5; BA.2.6; BA.2.7; BA.2.8; BA.2.10; BA.2.12. Antigen "B.1.1.529" includes: B.1.1.529; BA.1; BA.1.15. Source data are provided as a Source Data file.

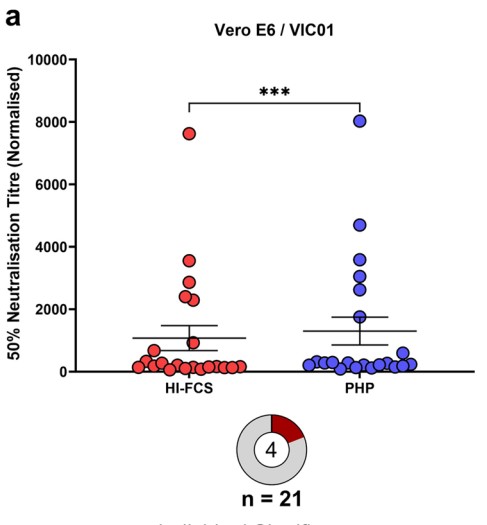

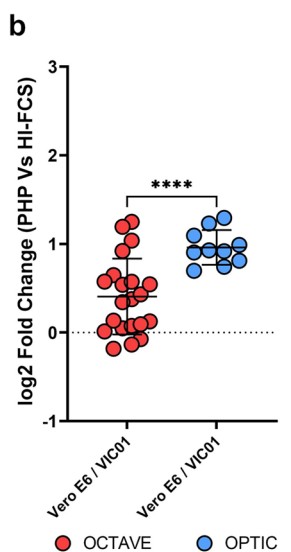

**Fig. 4 | Comparison of 50% neutralisation titres (NT50s) against SARS-CoV-2 (VIC01) in the OCTAVE and OPTIC cohorts. a** NT50s were determined via microneutralisation assays (MNAs) for all OCTAVE vaccinee serum samples (*n* = 21) supplemented with either heat-inactivated FCS (HI-FCS) or pooled human plasma (PHP). Each dot represents the duplicate NT50 values of a single sample from the first series of experiments and significance between populations was determined using a two-tailed Wilcoxon matched-pairs test in GraphPad Prism (Version 10) (*p* = 0.0003). Error bars show the mean value with the standard error. Significance between HI-FCS and PHP for each individual was determined by the sum-of-squares *F*-test with non-overlapping 95% confidence intervals and only the significant samples were repeated across 3 independent experiments. The pie chart shows the

number of these samples with a significant increase (central number represented in red) against the total population (outside number). **b** Log2 fold-change comparing the addition of PHP versus HI-FCS on NT50 values against SARS-CoV-2. Significance between Vero E6/VIC01 conditions for the NT50s of OCTAVE (*n* = 21) and OPTIC vaccinee serum samples (*n* = 10) were determined using a two-tailed Welch's *t*-test in GraphPad Prism (Version 10) (*p* = <0.0001). Each spot shows the difference in NT50 values between the addition of PHP or HI-FCS for each sample, determined via a 4-parameter logistic curve using 7 (OCTAVE) sera dilution points as described for (**a**) or using 12 (OPTIC) sera dilution points with four replicates across duplicate assays (OPTIC). Error bars show the standard deviation. *p < 0.05, **p < 0.01, ***p < 0.001, ****p < 0.0001. Source data are provided as a Source Data file.

Non-Enhanced cohort but not in the Enhanced cohort (Supplementary Fig. 6).

To understand which of these antibody characteristics might be important for the complement-mediated enhancement of neutralisation to occur, and if a combination of factors is required, we performed a supervised random forest (RF) machine learning algorithm and LASSO and ridge regression. The RF model classified whether a sample showed a significant complement-mediated enhancement of neutralisation with a mean accuracy of 77.5% and a 14.5% CV across 20 iterations. The model's ability to separate positive and negative cases across all classification thresholds as measured by the area under the curve (AUC) was 0.864 with 8.5% CV, suggesting it's fit for purpose (Supplementary Fig. 7). The most important feature for model accuracy (Supplementary Fig. 7) and node purity was IgG3 (Fig. 5f). We then performed LASSO and ridge regression analyses using the same

measurements of antibody characteristics as verification with a separate model. SARS-CoV-2 spike-specific IgG3 titres were again highlighted as the most important factor in both models and was positively associated with complement-enhanced neutralisation (Fig. 5g). LASSO regression highlighted four important variables (SARS-CoV-2 spike-specific IgG3 [COEFF 2.76], HKU1-spike-specific IgG [COEFF 1.00], SARS-CoV-2 spike-specific IgG1 [COEFF 0.43], SARS-CoV-1 spike-specific IgG [COEFF -0.17]) and ridge regression highlighted two (SARS-CoV-2 spike-specific IgG3 [COEFF 1.18] and HKU1 spike-specific IgG [COEFF 0.52]).

## Discussion

Live virus neutralisation assays are the gold standard for determining neutralising antibody titres, which have significant implications for understanding immunity and correlates of protection. Replenishment

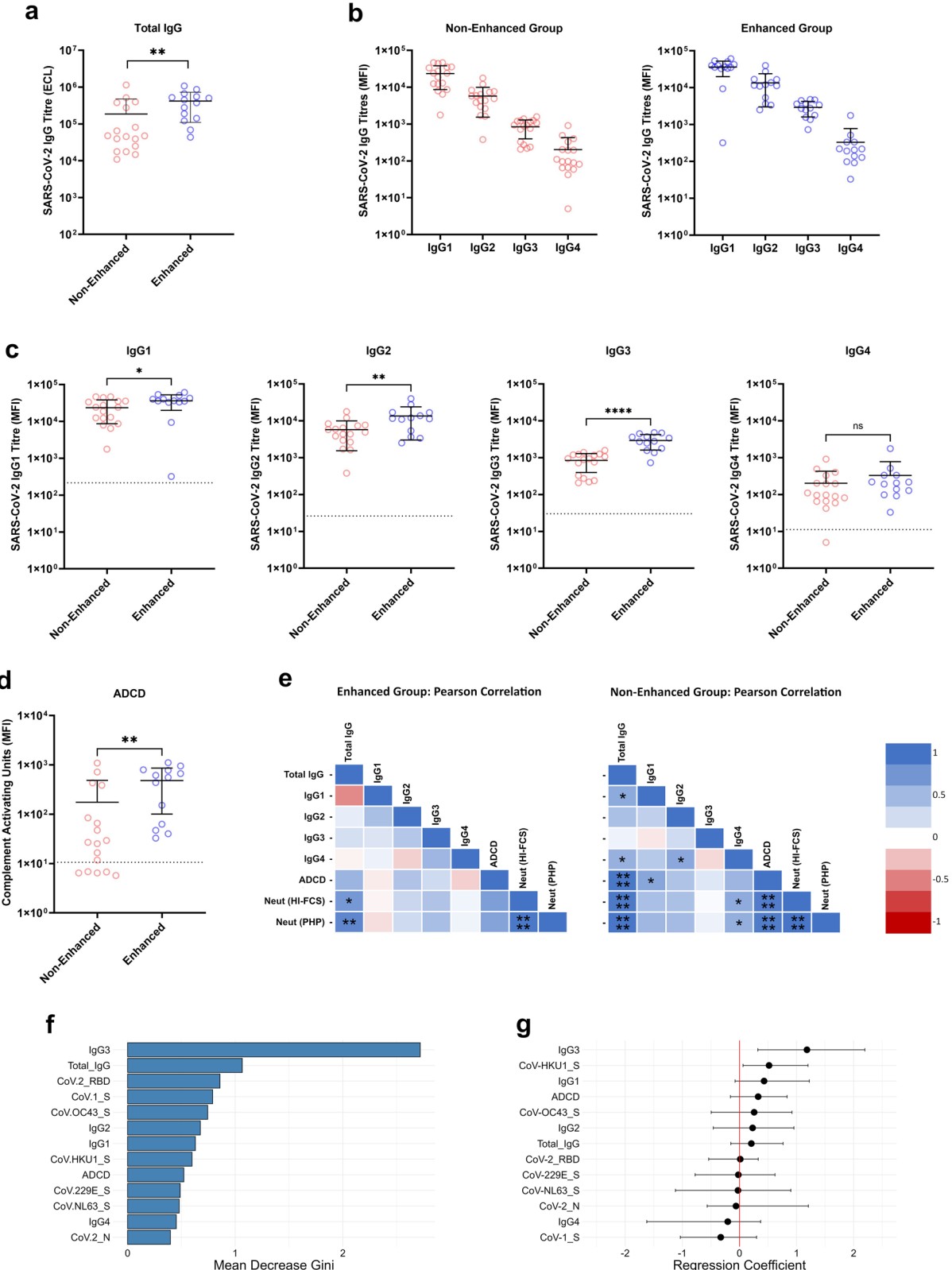

of the complement system in neutralisation assays can enhance neutralising antibody titres against a range of viruses[26] but this has not previously been shown for SARS-CoV-2. We have shown that replenishment of the complement system in live neutralisation assays through the addition of exogenous human plasma, significantly increases SARS-CoV-2 antibody neutralisation titres of vaccinee serum. The magnitude of this enhancement varies depending on cell lines,

viral strains, and the immune sera. This effect is heat labile, reduces ACE2 binding to Coronavirus spike antigens, and the complement protein C3 is required for maximum efficiency. Our collective evidence suggests that the complement system enhances SARS-CoV-2 neutralisation titres through the inhibition of cell attachment and entry.

The complement-mediated enhancement of NT50s was most profound when using Calu-3 cells compared to Vero E6 cells.

**Fig. 5 | Comparison of antibody characteristics between samples with (Enhanced cohort) or without (Non-Enhanced cohort) evidence of complement-enhanced neutralisation. a** Total SARS-CoV-2 spike IgG titres measured via electrochemiluminescence (ECL) using Meso Scale Discovery assays ($p = 0.0029$). Each dot represents the average ECL signal of each background-subtracted sample tested in duplicate (**b**) Median fluorescence intensity (MFI) of SARS-CoV-2 spike-specific IgG1-4 in all samples determined via flow cytometry. Each dot represents the average MFI of each background-subtracted sample tested in duplicate. **c** Pairwise comparison of MFI of IgG1-4 SARS-CoV-2 spike-specific titres (IgG1, $p = 0.0321$; IgG2, $p = 0.0098$; IgG3, $p = <0.0001$; IgG4, $p = 0.3216$). Each dot represents MFI values as described for (**b**) Dotted lines show the mean MFI of negative samples plus 3 standard deviations. **d** Pairwise comparison of antibody-dependent complement deposition (ADCD) between the Enhanced and Non-Enhanced cohorts, using MFI to measure C3c deposition ($p = 0.0031$). Each dot represents the average MFI from each sample tested in duplicate and interpolated from a standard curve assigned with arbitrary 'Complement Activating Units'. For (**a**–**d**), statistical significance was determined using an unpaired, two-sided *t*-test

and error bars show the mean value with standard deviation (SD), comparing the Enhanced (**a**, $n = 14$; **b**–**d**, $n = 13$) and Non-Enhanced cohorts ($n = 17$). **e** Two-tailed Pearson correlation with Benjamini Hochberg false discovery rate of 0.05 to compare relationships of antibody characteristics within the two cohorts. **f** Mean decrease in gini, representing the order of variable importance for determining node purity in the random forest model to classify outcome of complement-enhanced neutralisation. **g** Ridge regression coefficients in order of positive relationship with complement-enhanced neutralisation. Dots represent the mean coefficient for each antibody characteristic, with 95% confidence intervals (CIs). Features with CIs not overlapping 0 were considered to be important predictors. **f** and **g** used data containing total SARS-CoV-2 spike-specific IgG titres, IgG subclass titres, ADCD, and antibody epitope specificity to Coronavirus antigens to determine complement-enhanced neutralisation. Statistical analysis for (**a**–**e**) was determined using GraphPad Prism (Version 10). Modelling for (**f** and **g**) was performed in RStudio. \*$p < 0.05$, \*\*$p < 0.01$, \*\*\*$p < 0.001$, \*\*\*\*$p < 0.0001$. Receptor binding domain (RBD), spike protein (S), nucleocapsid (N), coronavirus (CoV). Source data are provided as a Source Data file.

---

Differences between cell lines could support the mechanistic explanation of reduced cell attachment/entry. For example, TMPRSS2 is a co-receptor for SARS-CoV-2 infection, which cleaves the spike protein at the polybasic cleavage site between S1 and S2[35]. Unlike Calu-3 cells, Vero E6 cells do not express TMPRSS2[36]. Therefore, antibodies which target certain epitopes involved in TMPRSS2 interactions may show greater enhancement with complement. However, the Omicron spike protein is inefficiently cleaved by TMPRSS2 and so this explanation alone would not explain why the greatest enhancement in neutralisation was observed against the BA.1 infection of Calu-3 cells. Also, we did not observe a significant overall difference in NT50s of the OPTIC cohort using Vero E6 cells compared to Vero E6 cells constitutively expressing TMPRSS2 (Supplementary Fig. 8). However, given the limited sample size due to serum limitations, it's possible that individual sample differences may be masked by the overall population.

ACE2 expression is also reportedly higher in Calu-3 cells compared to Vero E6 cells[37] and a reduction in ACE2 binding through complement activity (supported by our ACE2 competition data) could further support this theory. Similar differences between cell lines have been reported for HCMV neutralisation with complement[11]. Cell type-dependent neutralisation has also been reported for other viruses including influenza virus[38], HSV-1[39], and flaviviruses including West Nile virus and dengue virus[40]. For flaviviruses, their structural heterogeneity and how this corresponded to cell attachment and epitope availability for antibody binding resulted in differences in neutralisation titres. In this study, the enhancement of NT50s against BA.1 was significantly greater compared to VIC01 for Calu-3 cells, but not Vero E6 cells. Antibody epitope specificity against the viral strains and how this corresponds to the host-cell receptors might explain this observation. Also, cross-reactive antibodies such as those against BA.1 may have a greater dependency on complement for neutralisation. This is an important consideration for the continued emergence of new variants[41]. Whilst epidemiological data has demonstrated a loss of immunity with emerging Omicron variants[42,43], the levels of cross-protection may be higher than first evaluated by conventional neutralisation assays for some individuals where antibody binding is maintained. For example, preservation of antibody binding and Fc activity against full-length Omicron spike protein has been demonstrated despite loss of neutralisation and binding to the receptor binding domain[44].

The enhancement in neutralisation was shown to be heat labile, in accordance with complement activity, and occurred prior to contact with the cells or within 2 h of infection. The use of TEM, compstatin, and ACE2 competition assays further supported our hypothesis that the complement-mediated enhancement is due to improved inhibition of cell attachment/entry. Our observations via TEM showed an absence of any obvious viral aggregation and lysis; two of the most commonly

reported mechanisms for neutralisation enhancement that have been demonstrated for other viruses using TEM[5,45]. The use of compstatin showed that the complement protein C3 was required for the most efficient neutralisation but that it was not essential. This excludes the necessity of downstream proteins that would be involved in viral lysis (C5-C9) and suggests that a combination of proteins C1, C4, C2 and C3 may be sufficient to enhance neutralisation, as shown for other viruses[4,6,28,32,46]. Finally, the ACE2 competition assays showed a significant reduction in ACE2 binding for all SARS-CoV-2 spike variants tested following the addition of complement prior to heat inactivation. Interestingly, although the ACE2 competition assay may be used as a surrogate for conventional neutralisation assays[47], the reduction in ACE2 binding was modest compared to the increases in neutralisation, suggesting the reduced ACE2 binding may only be a partial mechanistic explanation.

The complement-mediated enhancement of NT50s varied significantly within and between the OPTIC and OCTAVE vaccinee serum cohorts. One explanation for this could be differences in antibody characteristics, which are known to influence complement activity, these include: epitope specificity, glycosylation, affinity, isotype, and IgG subclass[26]. We found significant increases in total SARS-CoV-2 spike-specific IgG, IgG1, IgG2, IgG3, ADCD, and differences in Coronavirus protein binding characteristics for samples that showed a complement-mediated enhancement of neutralisation. IgG4 has a mostly inert role in complement utilisation[26] and this constituted a higher proportion of total IgG in the Non-Enhanced group compared to the Enhanced group. Machine learning approaches with RF, and both LASSO and ridge regression models, highlighted IgG3 as the most important predictor from the observations tested, with a positive relationship to complement-enhanced neutralisation. We believe this further supports our findings and hypothesis that the mechanism is primarily mediated through the binding of proteins C1–C3, as IgG3 has the strongest affinity for C1q binding out of the IgG subclasses[48]. Future studies could expand on this using purified antibodies with the same epitope recognition, that differ in the IgG subclass. Similarly, to experimentally address the significance of epitopes, antibodies with the same IgG subclass that differ in epitope recognition could be used.

Whilst the use of serum samples from two cohorts provided insights into the different mechanics of complement-enhanced neutralisation, a single, matched cohort would be required to understand their relative impacts. Several differences between these two cohorts might have influenced their antibody profiles leading to these responses, including: health status (OCTAVE: immunocompromised participants; OPTIC: healthy participants), vaccine status (OCTAVE: ChAdOx1 Vaccine; OPTIC: COVID-19 mRNA vaccine BNT162b2 (Pfizer/BioNTech)), timing of sample collection post-boost (OCTAVE: 25–67 days; OPTIC: 7 days), age differences, sex differences, and the

persistence of IgM. A systematic approach using matched cohorts would be required to deconvolute these factors. Comparison of IgG binding to Coronavirus spike proteins and antibody characteristics including IgG titre, IgG subclass, and ADCD comparing the OPTIC and OCTAVE cohorts are shown in Supplementary Figs. 9 and 10, respectively.

The significant variation in the responses to complement between individuals could be an important consideration for correlates of protection. Whilst neutralisation assays are a good indicator of protection[49–51], they are not without limitations as some individuals may be unique in exhibiting complement-enhanced neutralisation or rely on other Fc-mediated antibody effector functions[52,53]. The inclusion of complement to neutralisation assays has been shown to strengthen correlations with protection in vivo for other viruses[3–5]. This could be especially important for antibodies which exhibit complement-dependent neutralisation, as shown for some samples within this study. Alternatively, given the large variation in enhancement for some donors and the complex heterogeneity of antibody characteristics involved in protection, the impact of complement in neutralisation assays and its relationship to protection in vivo may not be easily correlated[2]. Future SARS-CoV-2 protection studies could consider the evidence provided in this study to assess this possibility.

This study shows that the complement system can enhance SARS-CoV-2 neutralisation titres for some vaccinated individuals and that this mechanism is likely mediated through the inhibition of viral attachment/entry to the host cell. Antibodies which bind outside the epitope for receptor binding may be dependent on the subsequent binding and deposition of complement proteins to then mask these epitopes. This seems particularly important for cross-protection and the threat of emerging variants, where our results showed up to an 83-fold increase in neutralisation against BA.1 using vaccinated sera which pre-dates its emergence. Whilst this work does not include the most recent SARS-CoV-2 Omicron lineages, the immunological principles remain the same and were demonstrated for two viruses further apart in lineage than BA.1 to the currently circulating strains[41]. Given the large heterogeneity between the samples tested in this study, it's possible that multiple or alternate mechanisms may be responsible for the complement-enhanced neutralisation.

We believe these findings hold physiological relevance, as fully functional complement activity occurs in the lung lavage fluid of healthy individuals, albeit at a reduced capacity compared to serum (functional activity of the classical pathway was ~39% of the magnitude of serum activity)[54]. The PHP in this study was tested at 20% of physiological concentrations in serum. Furthermore, the complement proteins required for full functional activity can be collectively synthesised by various non-immune cells resident in the lung (alveolar type II epithelial cells[55–57], AT2 cells[57], club cells[57], fibroblasts[55,57], goblet cells[57], mesothelial cells[57], and mucous cells[57]) as well as immune cells capable of residing in the lung or migrating during infection (monocytes, macrophages, dendritic cells)[58]. The results from this study demonstrate that enhanced neutralisation requires the complement protein C3 for maximum efficiency and only partial enhancement is acquired in its absence. Therefore, only the synthesis of the upstream proteins C1, C4, and C2 may be required to enhance neutralisation. Lastly, whilst SARS-CoV-2 primarily infects cells within the respiratory tract, productive infection also occurs within cardiomyocytes[59], renal parenchymal cells[60], hepatocytes[61], neurons and glial cells[62], where the virus has been identified in the relevant organs in infected patients[63].

Overall, this study demonstrates the importance of the complement system in enhancing SARS-CoV-2 antibody neutralisation titres and explores the various underlying mechanisms. The complement-enhanced neutralisation varies in magnitude between individuals and demonstrates up to an ~83-fold increase in neutralisation with cross-reactive antibodies to SARS-CoV-2 strains. This mechanism has physiological relevance for SARS-CoV-2 infections and is likely mediated

by complement proteins C1–C3 with reduced ACE2-spike interactions. These findings should be considered when assessing future vaccine and therapeutic efficacies and their possible implications for correlates of protection.

## Methods

### Ethical approval and sample cohorts

Pooled human plasma (PHP) from five healthy UK donors was used as a source of complement and was collected as previously described[64] by the High Consequence Emerging Viruses Group at the University of Oxford. The PHP was collected in May 2021 and was confirmed negative for IgG antibody reactivity with the SARS-CoV-2 spike protein via ELISA. The PHP has been used in previous studies demonstrating its complement activity[19,65]. Written informed consent was obtained from all donors. For the antibody-dependent complement deposition assays (ADCD), IgG- and IgM-depleted human complement (Pel-Freeze Biologicals) was used.

The OCTAVE (Observational Cohort trial T cells, Antibodies and Vaccine Efficacy in SARS-CoV-2) trial (ISRCTN 12821688) aims to assess the SARS-CoV-2 vaccine responses of immunocompromised individuals that were part of the UK national COVID-19 vaccination programme and the majority of subjects received either the COVID-19 mRNA vaccine BNT162b2 (Pfizer/BioNTech) or the ChAdOx1 Vaccine (AstraZeneca formerly AZD1222)[33,66]. The serum from individuals in the OCTAVE cohort used within this study ($n = 21$) was collected 25–67 days post-boost with the ChAdOx1 Vaccine, between May and July 2021. These samples were randomly selected and the corresponding individuals had a diagnosis belonging to one of the following groups: autoimmune hepatitis ($n = 2$), liver cirrhosis (Child Pugh A ($n = 6$) or Child Pugh B ($n = 3$)), Crohn's disease ($n = 2$), ulcerative colitis ($n = 6$), kidney transplant ($n = 2$). All patients and participants provided their written informed consent to participate in this study. The OCTAVE Trial was approved by the UK Medicines and Healthcare Products Regulatory Agency (MHRA) on 5 February 2021 and by the London and Chelsea Research Ethics Committee (REC ref.: 21:/HRA/0489) on 12 February 2021. The protocol has subsequently been amended eight times with five substantial amendments (with ethical approvals dated 3 March 2021, 19 April 2021, 24 December 2021 and 4 April 2022) and three non-substantial amendments: protocol versions dated 22 April 2021, 14 July 2021 and 10 September 2021. The trial is registered on ISRCTN12821688.

The OPTIC (Oxford Protective T-cell Immunity to Coronavirus) study is a prospective, longitudinal observational cohort study of healthcare workers (HCWs) as part of the national PITCH (Protective Immunity from T Cells in Healthcare workers) consortium. HCWs defined as SARS-CoV-2 naïve based on documented PCR and/or serology results were recruited after vaccination with the COVID-19 mRNA Vaccine BNT162b2 (Pfizer)[34]. The serum from individuals in the OPTIC cohort used within this study ($n = 10$) was collected 7-days post-boost, in January 2021. The samples were randomly selected for use in this study. The OPTIC healthcare worker participants were recruited under the GI Biobank Study 16/YH/0247, approved by the research ethics committee (REC) at Yorkshire & The Humber - Sheffield Research Ethics Committee on 29 July 2016, which was amended for this purpose on 8 June 2020. All patients and participants provided their written informed consent to participate in this study.

All OPTIC and OCTAVE serum samples were heat-inactivated at 56 °C for 30 min prior to their use in this study.

### Cells and virus stocks

The wild-type Victoria/01/2020 (VIC01) isolate was originally supplied by the Doherty Centre Melbourne[67] and was passaged in Vero E6/TMPRSS2 cells (NIBSC Research Reagent Repository, UK. NIBSC reference 100978), and confirmed identical to GenBank MT007544.1, B hCoV-19_Australia_VIC01_2020_ EPI_ ISL_ 406844_ 2020-01-25. The

BA.1 Omicron/BA.1.1.529 strain (hCoV/England/FCI-099/2021) was originally provided by the Francis Crick Institute and subsequently by Professor William James from the Sir William Dunn School of Pathology, University of Oxford. The virus was passaged in Vero E6/TMPRSS2 cells (provided by the NIBSC Research Reagent Repository, UK. NIBSC Reference 100978) and confirmed identical to GenBank ON020748.1.

Vero E6 cells used within this study were obtained from the European Culture of Authenticated Cell Cultures (non-human primate kidney, Vero 76, clone E6, European Culture of Authenticated Cell Cultures, Salisbury, UK, 85020206) and Calu-3 cells were obtained from the American Type Culture Collection (human lung adenocarcinoma, ATCC, HTB-55).

### Microneutralisation assay

MNAs were modified from previous publications[68] to determine whether the addition of PHP as a source of complement, compared to HI-FCS or media-only, could enhance neutralisation titres. Vaccinee serum samples were serially diluted 1:2 across 12 (OPTIC cohort starting dilution: 1:80 against VIC01 or 1:10 against BA.1) or 7 (OCTAVE cohort starting dilution: 1:10 against VIC01) dilution points, in a volume of 10 μl per well. Then 10 μl of HI-FCS, PHP, or equivalent volumes of assay media, were added to each dilution point in duplicate, for a final concentration of 20%. SARS-CoV-2 VIC01 (OPTIC and OCTAVE cohorts) and BA.1 (OPTIC cohort) strains were added to each well at a final concentration of ≥100 FFU, for a final volume of 40 μl. The samples were then incubated for 1 h at 37 °C, 5% $CO_2$ for neutralisation to occur. For the infection of Vero E6 cells, the cells were prepared at a concentration of $4.5 \times 10^5$ cells/ml in 1% Vero E6 assay media (Gibco™ DMEM with 1% HI-FCS and 1% penicillin-streptomycin) and 100 μl was added to each well. For the infection of Calu-3 cells, 70,000 cells per well were pre-seeded for 24 h in 10% Calu-3 growth media (Gibco™ MEM, 10% HI-FCS, 1% penicillin-streptomycin, 1 mM sodium pyruvate, 2 mM L-glutamine, 1× non-essential amino acids) which were then washed in DPBS and replaced with 100 μl of 1% Calu-3 assay media (Gibco™ MEM, 1% HI-FCS, 1% penicillin-streptomycin, 1 mM sodium pyruvate, 2 mM L-glutamine, 1x non-essential amino acids). 35 μl of the virus/serum mixture was then transferred to the Calu-3 cell monolayer.

All samples were then incubated for 2 h at 37 °C, 5% $CO_2$. Finally, 100 μl of 1.5% carboxymethyl cellulose (CMC) in assay media was added to all samples and the plates were returned to the incubator at 37 °C, 5% $CO_2$ until 20 h post-infection (VIC01) or 24 h post-infection (BA.1). All samples were then developed according to the "Microneutralisation assay development" section. All samples were tested in duplicate and assays were performed in duplicate (OPTIC), or in triplicate following a significant result (OCTAVE).

### Microneutralisation assay modification 1: use of heat-inactivated PHP

The following modifications were made from the previously described MNA method to determine the effect of heat inactivation on PHP. All serum samples (OPTIC cohort, $n = 10$) were diluted in 1% Calu-3 or Vero E6 assay media to a final dilution of 1:1000, 1:1500, and 1:4500. PHP, or heat-inactivated PHP (HI-PHP) following a 30 min incubation at 56 °C, were added in triplicate for a final concentration of 20%. The SARS-CoV-2 VIC01 strain was added to each well at a final concentration of ≥100 FFU and the samples were incubated for 1 h at 37 °C, 5% $CO_2$. 35 μl of the virus-serum mixture was transferred to a Calu-3, Vero E6, or Vero E6 with TMPRSS2 cell monolayer and incubated for 2 h at 37 °C with 5% $CO_2$ before aspirating the media, washing the wells with 200 μl of DPBS, and replacing with 100 μl of fresh 1% Calu-3 or Vero E6 assay media. The assay then progressed as previously described for MNAs. All samples were tested in triplicate and assays were performed in duplicate.

### Microneutralisation assay modification 2: addition of compstatin

The following modifications were made from the previously described MNA method to determine the effect of compstatin on PHP. All serum samples (OPTIC cohort, $n = 3$) were diluted in 1% Calu-3 assay media to a final dilution of 1:1200 with the addition of either: 20% PHP with 130 μM compstatin (amino acid sequence: ICVVQDWGHHRCT-NH2), 20% PHP with 130 μM compstatin control peptide (amino acid sequence: IAVVQDWGHHRAT-NH2), or 1% Calu-3 assay media. The assay then progressed as previously described for MNAs. All samples were tested in triplicate and assays were performed in duplicate.

### Microneutralisation assay development

MNAs were prepared as previously described. After 20 h (VIC01) or 24 h (BA.1) post-infection, the CMC overlay was aspirated, each well was washed with 200 μl of DPBS, and the cells were fixed with 100 μl of 4% paraformaldehyde (PFA) in DPBS for 30 min. The 4% PFA was then aspirated and replaced with 100 μl of permeabilization buffer (2% Triton X-100 in DPBS). The plates were then incubated at 37 °C, 5% $CO_2$ for 30 min. All wells were then aspirated and washed three times with 100 μl of wash buffer (0.1% tween-20 in DPBS), and 50 μl of anti-SARS-CoV-2-nucleocapsid antibody (generously provided by Tiong Tan at the Radcliffe Department of Medicine, University of Oxford, UK) was diluted 1:5000 in wash buffer and added to each well. All samples were then incubated at RT for 1 h whilst shaking at 150 rpm, aspirated, and washed three times with 100 μl of wash buffer. Anti-human IgG antibody (Merck, #A0170-1ML, polyclonal) conjugated to peroxidase was then diluted 1:5000 in wash buffer, and 50 μl was added to each well. All samples were then incubated at RT for 1 h whilst shaking at 150 rpm, aspirated, and washed three times with 100 μl of wash buffer. The samples were developed using 40 μl of TrueBlue™ Peroxidase Substrate (Seracare) in each well and incubated for 10 min at RT whilst shaking at 150 rpm. The substrate was aspirated and 100 μl of ultrapure water was added to each well before a 5 min incubation at RT, whilst shaking at 150 rpm. Finally, all samples were aspirated and left to dry at RT for 45 min before determining the number of foci with the ImmunoSpot® (Cellular Technology LTD). Foci were automatically counted using the BioSpot™ Software Suite and subjected to quality control to verify the counting accuracy and ensure integrity of the cell monolayer.

### Transmission electron microscopy

Transmission electron microscopy (TEM) was used to determine possible aggregation and/or lysis of SARS-CoV-2 virus particles. OPTIC samples 8 and 10 were diluted in 1% Vero E6 assay media to a final dilution of 1:1200 with either 20% PHP or 20% HI-PHP. The samples were then incubated for 1 h at 37 °C with 150 FFU of SARS-CoV-2 (VIC01) per sample and inactivated in a final concentration of 4% PFA in DPBS for 30 min. Negative staining for TEM was prepared as follows: 300 mesh carbon support film coated copper grids were glow discharged for 25 s using a Pelco easiGlow Glow Discharging Unit. Grids were then placed on a 10 μl droplet of the sample and incubated for 2 min at RT, followed by removal of excess sample using Whatman No1 filter paper and staining with 2% uranyl acetate for 10 s. Excess uranyl acetate was removed using Whatman No1 filter paper. The samples were allowed to air dry and then analysed using a Jeol 1400 TEM with a Gatan Rio CMOS detector. A total of 136 images were acquired at ×5000, ×10,000 and ×20,000 magnifications.

### Meso scale discovery ACE2 competition assay

Inhibition of SARS-CoV-2 antigen binding to recombinant human ACE2 protein by immune sera was measured using a multiplexed Meso Scale Discovery (MSD) immunoassay: SARS-CoV-2 Key Variant Spike Plate 1. Each well was coated with SARS-CoV-2 nucleocapsid (Wuhan) and spike antigens from the following lineages: Wuhan, Alpha, Beta, Delta,

Omicron. Full details can be found in Supplementary Table 1. MSD ACE2 competition assays were performed in 96-well plates with an initial blocking step using 150 μl of MSD Blocker A for 30 min at RT, whilst shaking at 500 rpm. The wells were then washed three times with 150 μl of 1× MSD Wash Buffer and 25 μl of immune sera at a final 1:10 or 1:100 dilution with either 20% PHP or 20% HI-PHP was added in duplicate. PHP and HI-PHP at a final concentration of 20%, without the addition of immune sera, were included to determine background signal. Following a 1 h incubation at 37 °C with shaking at 500 rpm, 25 μl of 1× recombinant SULFO-TAG Human ACE2 Protein was added and the plate was incubated at RT for 1 h with shaking at 500 rpm. The plates were then washed, 150 μl of MSD GOLD Read Buffer B was added to each well, and the electrochemiluminescence (ECL) signals were determined according to the manufacturer's instructions.

## SARS-CoV-2 spike conjugation to fluorescent beads

To determine the IgG subclasses and levels of ADCD of the OPTIC and OCTAVE serum samples, APC-fluorescent beads conjugated to the SARS-CoV-2 whole spike protein were used as previously described[52,53,69]. 500 μl of SPHERO™ Magnetic Flow Cytometry Multiplex Bead Assay particles (Spherotech) were pelleted using the EasyEights™ EasySep™ Magnet (STEMCELL Technologies), washed in 82 mM sodium phosphate buffer (pH 6.2), and activated in the same buffer containing 1.24 mg of N-hydroxysulfosuccinimide and 1-ethyl-3-[3-dimethlyyaminopropyl]carbodiimide-HCl for 20 min. The beads were then pelleted and washed twice in coupling buffer (50 mM 2-(N-morpholino) ethanesulfonic acid, pH 5.0) and resuspended in 240 μl of coupling buffer containing 14.5 μg of SARS-CoV-2 spike protein from the ancestral strain (Lake Pharma, 46328) for 2 h on a rotational mixer. The conjugated beads were then pelleted and washed twice in blocking buffer (PBS containing 2% BSA and 0.05% sodium azide, pH 7.4) and resuspended in 200 μl of the same buffer overnight on a rotational mixer. The beads were then pelleted, washed and resuspended in 500 μl of PBS containing 0.05% sodium azide, and stored at 4 °C until use.

## IgG subclass assay

SARS-CoV-2 spike-conjugated, magnetic, fluorescent beads were prepared as previously described and used to determine the levels of IgG1, IgG2, IgG3, and IgG4 in the OPTIC ($n = 9$) and OCTAVE ($n = 21$) serum samples. The beads were diluted to a concentration of 50 beads/μl and 20 μl was added to each well, with 30 μl of PBS containing heat-inactivated serum at a final dilution of 1:50, conducted in duplicate. The beads and serum were incubated for 1 h at RT whilst shaking at 700 rpm, then washed twice in 100 μl of wash buffer (PBS containing 0.1% tween-20) and resuspended in 100 μl of 1 μg/ml PE-conjugated IgG1 (Cambridge Bioscience, #9052-09, clone 4E3), IgG2 (Cambridge Bioscience, #9060-09, clone 31-7-4), IgG3 (Cambridge Bioscience, #9210-09, clone HP6050), or IgG4 (Cambridge Bioscience, #9200-09, clone HP6025) antibody in PBS. The samples were incubated for 1 h at RT whilst shaking at 700 rpm and then washed twice in 100 μl of wash buffer before resuspending in 50 μl of PBS. Duplicate samples were combined and a minimum of 100 beads per sample were acquired on the BD LSRFortessa X-20 flow cytometer. The median fluorescence intensity (MFI) of PE was determined using FlowJo (version 10.10.0) with the gating strategy shown in Supplementary Fig. 1. The background fluorescence for each serum sample was measured in the absence of secondary antibody and subtracted from the raw MFI values. A quality control sample was included in each experiment for all IgG subclasses to ensure reproducibility. A SARS-CoV-2 IgG negative sample was also included in each experiment for all IgG subclasses where the mean MFI plus three standard deviations across all replicates was used to determine the limit of detection.

## Antibody-dependent complement deposition assay

SARS-CoV-2 spike-conjugated, magnetic, fluorescent beads were prepared as previously described and used to determine the levels of ADCD in the OPTIC ($n = 9$) and OCTAVE ($n = 21$) serum samples. The beads were diluted to a concentration of 50 beads/μl and 25 μl was added to each well, with 25 μl of HBSS containing heat-inactivated serum at a final dilution of 1:100 or 1:500, conducted in duplicate. The beads and serum were incubated for 30 min at RT whilst shaking at 700 rpm, then washed twice in 100 μl of wash buffer (PBS containing 0.1% tween-20) and resuspended in 50 μl of HBSS with 10% IgG- and IgM-depleted human complement (Pel-Freeze Biologicals). The samples were incubated for 20 min at 37 °C whilst shaking at 700 rpm, then washed twice in 100 μl of wash buffer, and resuspended in 100 μl of FITC-conjugated C3c antibody (Abcam, #ab4212, polyclonal) diluted 1:500 in HBSS. The samples were incubated for 20 min at RT whilst shaking at 700 rpm, washed twice in 100 μl of wash buffer, and resuspended in 50 μl of HBSS. Duplicate samples were combined and a minimum of 100 beads per sample were acquired on the BD LSRFortessa X-20 flow cytometer. The MFI of FITC was determined using FlowJo (version 10.10.0) with the gating strategy shown in Supplementary Fig. 2. MFI values were interpolated from a standard curve using 4-parameter logistic regression and then multiplied by the difference in dilution factor. Standard curves were included in each experiment and the interpolated values were presented as arbitrary "complement activating units".

## Statistical analysis

All statistical analyses were performed in GraphPad Prism (Version 10) where $p < 0.05$ was considered significant. Normality tests were performed on all samples prior to analysis. Random forest and logistic regression with LASSO and ridge methods were performed in R/R Studio (version 4.4.1).

For the MNAs supplemented with either PHP, HI-FCS, or assay media-only, all samples were normalised using the respective cell-only or no-serum controls on each 96-well plate. The 50% neutralisation titres (NT50) were calculated using values from two (OPTIC cohort) or three (OCTAVE) independent experiments. Where an NT50 value could not be determined, an arbitrary value of 10 was assigned. Significant changes in NT50s for a single sample were determined using the sum-of-squares $F$-test with non-overlapping 95% confidence intervals (CIs). Population differences were determined using a one-way ANOVA (Tukey's multiple comparisons test) for the OPTIC cohort or a Wilcoxon matched-pairs test to compare the OPTIC and OCTAVE cohorts. Significant differences for the log2 fold-change in neutralisation between HI-FCS and PHP were determined using a one-way ANOVA mixed effects analysis with Geisser-Greenhouse correction and Šídák's multiple comparisons test.

For the MNAs comparing the effects of PHP and HI-PHP, multiple paired, two-sided $T$-tests were used to determine significance for each vaccinee serum sample. For the MNAs comparing effects with media-only, compstatin, or the control peptide, a two-way ANOVA with Tukey's multiple comparisons test was used.

For the MSD ACE2 competition assay, percentage inhibition was first calculated using the following formula according to the manufacturer's instructions: % Inhibition = (1 - (Average ECL Signal of Sample / Average ECL Signal of Diluent Only)) × 100. The background signal with 20% PHP or HI-PHP in absence of immune sera was then subtracted from the respective wells. Differences with the addition of PHP or HI-PHP were then determined using multiple paired, two-sided $T$-tests for each antigen.

Pearson correlations were performed using GraphPad Prism after confirmation of a normal Gaussian distribution and significance was determined using a Benjamini Hochberg false discovery rate of 0.05 to account for multiple testing.

Random forest was performed using the R package 'randomForest' with 500 trees and 4 variables at each split. The dataset was split 70/30 with training and test data respectively, and performed across 20 iterations. The Least Absolute Shrinkage and Selection Operator (LASSO) and ridge logistic regression were performed using the R package 'glmnet', with alpha = 1 (LASSO) or alpha = 0 (ridge) and the optimal regularisation parameter lambda was determined through 10-fold cross-validation. This was followed by bootstrapping ($B$ = 1000) to generate 95% confidence intervals and variables with CIs non-overlapping zero were considered important. Important variables for LASSO regression were non-zero coefficients.

### Reporting summary
Further information on research design is available in the Nature Portfolio Reporting Summary linked to this article.

### Data availability
The accession codes referenced in this study were previously deposited in GenBank under the accession codes MT007544.1 [https://www.ncbi.nlm.nih.gov/nuccore/MT007544.1] (identical to SARS-CoV-2 VIC01 strain in this study) and ON020748.1 [https://www.ncbi.nlm.nih.gov/nuccore/ON020748.1] (identical to SARS-CoV-2 BA.1 strain in this study). The authors declare that the data supporting the findings of this study are available within the paper and its supplementary information files. All data generated within this study is provided in the Source Data file. Source data are provided with this paper.

### Code availability
Sample code and input data for the analyses in this study are available on GitHub: https://github.com/jmellors/Complement-Mediated-Enhancement-of-SARS-CoV-2-Antibody-Neutralisation-Potency. https://doi.org/10.5281/zenodo.14548585.

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

## Acknowledgements

This work was supported by the US Food and Drug Administration Medical Countermeasures Initiative contract 75F40120C00085 awarded to M.C. and J.H. The OCTAVE trial was sponsored by the University

of Birmingham and supported by the National Core Studies Immunology (NCSi) program and funded by a grant from UK Research and Innovation (UKRI) administered by the Medical Research Council (grant reference number MC_PC_20031). It was designated an Urgent Public Health study by the National Institute of Health Research (NIHR). The trial was designed, initiated and conducted independently by the trial investigators and delivered by the CRCTU, University of Birmingham. The OPTIC study was supported by the Department of Health and Social Care and by UKRI/MRC (MR/W02067X/1 and MR/X009297/1) as part of the PITCH (Protective Immunity from T cells to Covid-19 in Health workers) consortium. P.K. and E.B. are NIHR Senior Investigators, and P.K. is funded by WT222426/Z/21/Z and NIH (U19 I082360). S.J.D. is funded by an NIHR Global Research Professorship (NIHR300791). The OCTAVE trial was sponsored by the University of Birmingham and supported by the National Core Studies Immunology (NCSi) program. The trial was designed, initiated and conducted independently by the trial investigators and delivered by the CRCTU, University of Birmingham. We would like to acknowledge Dr Benjamin Wright from the Centre for Human Genetics, University of Oxford for their bioinformatics support.

## Author contributions

J.M. conducted the experiments, analysed the data and drafted the manuscript. J.M. and R.D. conducted the transmission electron microscopy experiments and wrote the relevant methods section. R.D. provided technical expertise on the interpretations of transmission electron microscopy. J.M., T.T. and S.L. collected the pooled human plasma and assessed its suitability for use in the study. OCTAVE consortium, OPTIC consortium, E.B., S.J.D. and P.K. established the clinical cohorts and collected and processed the clinical samples. J.M., M.C. and J.H. conceptualised the project and M.C. provided technical and intellectual guidance and supervised this work. All authors reviewed the manuscript and contributed to the final draft.

## Competing interests

The authors declare no competing interests.

## Additional information

## OCTAVE Consortium

Iain McInnes[4], Stefan Siebert[4], Pam Kearns[4], Dan Rea[4], Gordon Cook[4], Michelle Willicombe[4], David Thomas[4], Eleanor Barnes[4,5], Thushan de Silva[4], Lucy Wedderburn[4], Rossa Brugha[4], Jessica Bate[4], Carl Goodyear[4], Alex Richter[4], John Snowden[4], Jack Satsangi[4], Sean Hua Lim[4], Amanda Kirkham[4], Sarah Bowden[4], Sophia Magwaro[4], Ana Hughes[4], Ann Pope[4], Elspeth Insch[4], Vicky Churchill[4], Richard Beesley[4] & Eilean MacDonald[4]

## OPTIC Consortium

Eleanor Barnes[4,5], Sagida Bibi[8], Miles Carroll ®[1] ✉, Christopher P. Conlon[9], Alexandra S. Deeks[9], Christina Dold[8], Susanna Dunachie[4,6], John Frater[9], Katie Jeffery[10], Paul Klenerman[4,5], Barbara Kronsteiner[9], Teresa Lambe[8], Stephanie Longet[3,9], Alexander J. Mentzer[9], Donal Skelly[11] & Lizzie Stafford[9]

[8]Oxford Vaccine Group, Department of Paediatrics, University of Oxford, Oxford, UK. [9]Nuffield Department of Medicine, University of Oxford, Oxford, UK. [10]Radcliffe Department of Medicine, University of Oxford, Oxford, UK. [11]Nuffield Department of Clinical Neurosciences, University of Oxford, Oxford, UK.

