## [Transparent Peer Review file · Nature Communications]

Complement-Mediated Enhancement of SARS-CoV-2 Antibody Neutralisation Potency in Vaccinated Individuals

Corresponding Author: Dr Jack Mellors

Version 0:

Reviewer comments:

Reviewer #1

(Remarks to the Author)

This manuscript is a thorough evaluation of the role of complement when it is included in serum antibody neutralization assays of SARS-CoV-2 in vitro. Complement is not generally included in such assays. The authors show that complement can be important but that its importance varies widely depending on virus, target cell and donor serum. This is an important conclusion. Other important conclusions include the demonstration that complement's main role is probably in enhancing the viral entry blocking capability of antibody and the complement C3 is critically important. They also show that the effects of complement are greatest for the SARS-CoV-2 Omicron variant that is least well neutralized by the serum antibodies.

Given that the correlation between neutralization and protection against SARS-CoV-2 is not good but not perfect, a possible hypothesis could be that the limited breakdown reflects the absence of complement in classical neutralization assays. The authors do not do protection studies to evaluate this possibility, which would probably be needed if the authors wish to make this case. However, the large variation they see according to donor etc suggest that it is unlikely that including complement would lead to a much better correlation between in vitro assay and in vivo protection. I feel the authors skirt around this problem a little but their study is strong on purely basic science grounds without a clear translational message for vaccine evaluation. The authors could look at this review:

<https://doi.org/10.1038/s41577-023-00858-w>

which concludes more generally that there is much variation in effector function activity of antibodies against viruses according to some of the parameters they note for a variety of potential reasons but which will make in vitro assays including effector function difficult to easily correlate generally with in vivo activity.

The authors might also include some additional references:

[doi:10.1016/j.chom.2009.09.003](https://doi.org/10.1016/j.chom.2009.09.003).

<https://www.nature.com/articles/s41541-017-0038-0>

[doi:10.1128/JVI.03690-13](https://doi.org/10.1128/JVI.03690-13)

Reviewer #2

(Remarks to the Author)

This report focuses on measuring the effect of complement on SARS-CoV-2 neutralization. The authors find that plasma, because of its complement activity, increases neutralization when added to immune serum. Complement enhances neutralization against two different strains of SARS-CoV-2 and in experiments using two different target cells, although both the virus strain and the type of target cell can modulate the degree or frequency of enhancement. The authors also conclude that complement proteins C1-C3 are important mediators of enhanced neutralization and that inhibition of interactions between S protein and ACE-2 are most responsible.

A major problem with the study is that only serum, rather than purified immunoglobulin was used for all the experiments. This

leaves open the possibility that serum components other than immunoglobulin were interacting with complement (or interacting with other components of the added plasma).

The comparisons of neutralizing activity between the different subject cohorts is not very informative given the differences between the cohorts. Some of these differences are pointed out by the authors, but a study of a single, well-defined cohort would be adequate for key conclusions reached.

The description of the microneutralization assay should clarify the volume and final concentration of serum. Were different volumes and concentrations used for the different cohorts?

Why was a different amount of virus used for the assays with HI-PHP (modification 1)?

The possibility of complement contributing to cell lysis was not explored, calling into question the conclusion that blocking of virus-host receptor interactions is largely responsible for enhanced neutralization.

For statistical analyses and calculation of standard errors, were duplicate samples included? In other words, if 6 replicates were done across two assays, did $N = 6? 12? 2?$ Since the replicates would not be independent, it seems that N should equal 2 for the scenario described—unless repeated measures were factored in.

Reviewer #3

(Remarks to the Author)

This study shows that as is the case for other viral diseases, replenishment of the complement system significantly increases SARS-CoV-2 antibody neutralisation titres of vaccinee serum. This study is important as it suggests that heat inactivation of plasma samples underestimates the neutralization capacity of antibodies against SARS-CoV-2. The observation that this varies with cell type viral strains, and the immune sera, also points to mechanism but this is not adequately addressed by this study. Theories are provided but in my opinion need to be experimentally explored in order to warrant publication.

Major points:

1. Experimental evidence should be provided to understand why different cells show different outcomes ie blocking of TMPRSS2, and why different cohorts exhibit different levels of complement mediated neutralization through depletion or measurement of IgG3 for example.
2. It is not clear why the authors insist that complement enhanced neutralization is somehow separate to Fc effector function as this is the region that complement would bind to on antibodies and one of their proposed explanation for differences between the cohorts are all the mechanisms that define Fc function. Using plasma samples in this study the authors should perform ADCD and note if there are any correlations. This may just be a description of the ability of the antibodies to deposit complement.

Minor queries

1. Worth looking at <https://www.nature.com/articles/s41591-024-03131-2> for the correlates of protection description
2. I find the extensive explanation of all the complement pathways unnecessary but feel the 3 mechanisms of complement enhanced neutralization underexplained.
3. Can the authors show the ELISA validation of the “healthy” complement showing that these individuals were not infected with SARS-CoV-2? Was nucleocapsid also run?
4. How were the dilutions for the PHP decided?
5. As neutralization is highly specific to samples, data presented in figure 2 should be shown in a table or heatmap where all samples are individually indicated or fold changes shown in the plot.
6. Can the authors show neutralization assay in presence of PHP alone? IE no antibody/sample
7. Does the 4500 dilution and the 1500 dilution correlate in Figure 3? Because several sample look discordant in their differences.
8. The authors show that the TEM did not significantly differ, however it is unclear how they make that assessment. Are multiple views somehow quantified in an unbiased way?
9. It seems that the mechanism of influence of complement is not clear cut, with only a limited decrease seen in ACE2 binding. Are their alternate mechanisms previously undiscovered that may explain this?

Version 1:

Reviewer comments:

Reviewer #1

(Remarks to the Author)

The authors have responded constructively to the reviewer comments to give an improved manuscript.

Reviewer #2

(Remarks to the Author)

How sensitive is the BioSpot software for determining the effects of antibody and complement on cells?

Reviewer #3

(Remarks to the Author)

I think the additional experiments and rewording improve the study and now warrant publication.

We thank the reviewers for their detailed comments on the manuscript and we have addressed their points below. Please note that line numbers correspond to the revised manuscript with tracked changes.

REVIEWER COMMENTS

Reviewer #1 (Remarks to the Author):

This manuscript is a thorough evaluation of the role of complement when it is included in serum antibody neutralization assays of SARS-CoV-2 in vitro. Complement is not generally included in such assays. The authors show that complement can be important but that its importance varies widely depending on virus, target cell and donor serum. This is an important conclusion. Other important conclusions include the demonstration that complement's main role is probably in enhancing the viral entry blocking capability of antibody and the complement C3 is critically important. They also show that the effects of complement are greatest for the SARS-CoV-2 Omicron variant that is least well neutralized by the serum antibodies.

Given that the correlation between neutralization and protection against SARS-CoV-2 is not good but not perfect, a possible hypothesis could be that the limited breakdown reflects the absence of complement in classical neutralization assays. The authors do not do protection studies to evaluate this possibility, which would probably be needed if the authors wish to make this case. However, the large variation they see according to donor etc suggest that it is unlikely that including complement would lead to a much better correlation between in vitro assay and in vivo protection. I feel the authors skirt around this problem a little but their study is strong on purely basic science grounds without a clear translational message for vaccine evaluation.

The authors could look at this review:

<https://doi.org/10.1038/s41577-023-00858-w>

which concludes more generally that there is much variation in effector function activity of antibodies against viruses according to some of the parameters they note for a variety of potential reasons but which will make in vitro assays including effector function difficult to easily correlate generally with in vivo activity.

Thank you for your comments. We firstly agree that to make the case that the addition of complement to neutralisation assays would strengthen in vivo correlations, subsequent protection studies would be required to confirm this. We hope that future research conducting protection studies might consider this possibility and utilise what we have found in our study, but we do not intend to give the impression that we conclude this based on our current findings. We recognise that the reviewer raises an important point regarding the heterogeneity of samples in

response to complement and the possible implications of this for correlations. We interpret these findings as having the potential to strengthen correlations as donors that greatly benefit from the addition of complement, particularly those which show complete dependency on complement for neutralisation, may be anomalies in conventional neutralisation assays that better fit the trend when complement is present. However, we do accept another possibility is that some of the large variability we observe may not strengthen correlations. To address these points, we have now included the following:

Line 875 - 882:

“This could be especially important for antibodies which exhibit complement-dependent neutralisation, as shown for some samples within this study. Alternatively, given the large variation in enhancement for some donors and the complex heterogeneity of antibody characteristics involved in protection, the impact of complement in neutralisation assays and its relationship to protection in vivo may not be easily correlated². Future SARS-CoV-2 protection studies could consider the evidence provided in this study to assess this possibility.”

To further address the reviewer’s point surrounding the need for in vivo work to confirm the potential impact of complement-mediated neutralisation on correlates of protection, we have also amended the final conclusion, reducing the emphasis on correlates of protection:

Lines 919 – 933:

“Overall, this study demonstrates the importance of the complement system in enhancing SARS-CoV-2 antibody neutralisation titres and explores the various underlying mechanisms. The complement-enhanced neutralisation varies in magnitude between individuals and demonstrates up to an ~83-fold increase in neutralisation with cross-reactive antibodies to SARS-CoV-2 strains. This mechanism has physiological relevance for SARS-CoV-2 infections and is likely mediated by complement proteins C1-C3 with reduced ACE2-spike interactions. These findings should be considered when assessing future vaccine and therapeutic efficacies and their possible implications for correlates of protection.”

We found the suggested review to be very informative and helpful in shaping our discussion points and have now referenced this within the manuscript.

The authors might also include some additional references:

doi:10.1016/j.chom.2009.09.003.

<https://www.nature.com/articles/s41541-017-0038-0>

doi:10.1128/JVI.03690-13

Thank you for the suggestions, we found these to be very insightful and have included them within the manuscript.

Reviewer #2 (Remarks to the Author):

This report focuses on measuring the effect of complement on SARS-CoV-2 neutralization. The authors find that plasma, because of its complement activity, increases neutralization when added to immune serum. Complement enhances neutralization against two different strains of SARS-CoV-2 and in experiments using two different target cells, although both the virus strain and the type of target cell can modulate the degree or frequency of enhancement. The authors also conclude that complement proteins C1-C3 are important mediators of enhanced neutralization and that inhibition of interactions between S protein and ACE-2 are most responsible.

A major problem with the study is that only serum, rather than purified immunoglobulin was used for all the experiments. This leaves open the possibility that serum components other than immunoglobulin were interacting with complement (or interacting with other components of the added plasma).

Thank you for your comments. We believe that we can fully address these concerns through the following explanations.

To firstly address concerns that serum components other than antibody/immunoglobulin may be interacting with the complement system. The fundamental principle of the classical pathway is that it is antibody mediated. One exception to this rule is that the C1q protein can sometimes bind directly to a viral antigen without the need for antibody. In our study, as we do not see any neutralisation in the absence of immune sera, we can assume this interaction does not occur to any capacity that would confound our results. It is also possible for proteins such as pentraxin to be substituted for C1q and activate the classical pathway, but this still requires the presence of antibodies. Furthermore, as we do not see any neutralisation in the complement-only conditions, this rules out the possibility of the lectin and alternative pathways (which are antibody-independent) being responsible for these findings. To develop this point further, some serum samples were unable to neutralise the BA.1 strain, to which the antibodies were cross-reactive, irrespective of whether complement was present. This shows that when there is a loss of antibody binding in these samples, neutralisation cannot be recovered when adding complement, therefore supporting the notion that this is antibody-mediated. In summary, the complement activity was entirely dependent on the addition of immune sera which rules out the possibility of the lectin and alternative pathways, or the direct binding of C1q, to a capacity that would influence neutralisation. When there was a loss in cross-reactive antibody-binding of the serum to the BA.1 strain, in some instances, the neutralisation activity could not be recovered with the addition of complement. Thus, both antibody binding and the presence of the complement system were required for enhancement to occur.

To address the second part of the concern that the activity is not complement mediated and is rather another component of the plasma. We firstly show that the

effects of complement-mediated enhancement are heat-labile, which is characteristic of the complement system. This narrows down the possibilities that it is at least a heat-labile component of plasma that is responsible for the activity. Our evidence that complement is at the very least one of the main drivers in this effect is that with the use of compstatin – a complement component C3 inhibitor – we see a significant decline in the levels of neutralisation. This shows that the C3 protein, which is one of the central components in the pathway, is at least partially responsible for the enhancement. We previously explained how this was antibody-dependent, classical pathway mediated, and now dependent on C3 cleavage, which means that cleavage of proteins C1, C4, C2 would also be required for this activity. We have now conducted additional experiments which show antibody-dependent complement deposition of all samples in this study, measuring C3c deposition, which further shows that we are seeing complement activity which is antibody-mediated.

To summarise these points, we believe that all the evidence we have generated, together with the extensive literature that more broadly discusses this phenomenon and what is known about the complement system, supports the conclusion that enhanced neutralisation is antibody-dependent and complement mediated. As mentioned, we have conducted additional experiments measuring antibody subclass and ADCD which we hope will further alleviate the reviewer's concerns.

Whilst the use of purified antibodies could yield additional benefits towards understanding the impact of epitope or isotype, we believe that an extensive evaluation of this exceeds the scope of this study which has already taken many steps towards identifying the effects of complement on neutralisation, understanding the underlying mechanism, and then characterising these responses. The foundational work established in this study would help to guide future research in understanding more antibody factors that are important for complement-enhanced neutralisation. Given the broad heterogeneity between individuals, we believe a separate study aiming to address this particular question would be required.

We also believe that there is added value in studying immune sera, as this is most representative physiologically and when evaluating immune responses to vaccination and infection. To summarise this discussion point and the potential added value from using purified antibodies and how to approach this, we have now included the following in the discussion section: Lines 841 – 844:

“Future studies could expand on this using purified antibodies with the same epitope recognition, that differ in the IgG subclass. Similarly, to experimentally address the significance of epitopes, antibodies with the same IgG subclass that differ in epitope recognition could be used.”

The comparisons of neutralizing activity between the different subject cohorts is not very informative given the differences between the cohorts. Some of these differences are pointed out by the authors, but a study of a single, well-defined cohort would be adequate for key conclusions reached.

We recognise that the multiple differences between these cohorts limits the conclusions that we can make. However, we believe that the conclusions we have

made – which was the observation that complement-mediated enhancement of neutralisation significantly differed between the two cohorts – were suitable and within these limitations. We speculate that the differences between, and also within, these cohorts could be due to antibody epitope specificity, antibody subclass etc which are all reasonable speculations on why the responses differ. We have now conducted additional experiments to explore these questions. What we cannot conclude is the relative impact of these differences such as vaccination status and we believe we are clear in this fact, in that we do not make such conclusions.

Firstly, we stated that “Several differences between these two cohorts might have influenced their responses, including: health status (OCTAVE: immunocompromised participants; OPTIC: healthy participants), vaccine status (OCTAVE: ChAdOx1 Vaccine; OPTIC: COVID-19 mRNA vaccine BNT162b2 (Pfizer/BioNTech)), timing of sample collection post-boost (OCTAVE: 25 – 67 days; OPTIC: 7 days), age differences, sex differences, and the persistence of IgM.” We went on to state that “A systematic approach to matching samples would be required to deconvolute these factors”, which we believe already aligns with the reviewers comment that a single, well-defined cohort would be required to conclude differences. To clarify this stance and in response to the additional experiments, the relevant sections now read:

In the results section on lines 646 – 649:

“It is unclear which differences between the OPTIC and OCTAVE cohorts may be responsible for this, as the cohorts were not matched on factors including health status, vaccine status, time of sample collection, age, or sex.”

In the discussion section on lines 852 – 864:

“Whilst the use of serum samples from two cohorts provided insights into the different mechanics of complement-enhanced neutralisation, a single, matched cohort would be required to understand their relative impacts. Several differences between these two cohorts might have influenced their antibody profiles leading to these responses, including: health status (OCTAVE: immunocompromised participants; OPTIC: healthy participants), vaccine status (OCTAVE: ChAdOx1 Vaccine; OPTIC: COVID-19 mRNA vaccine BNT162b2 (Pfizer/BioNTech)), timing of sample collection post-boost (OCTAVE: 25 – 67 days; OPTIC: 7 days), age differences, sex differences, and the persistence of IgM. A systematic approach using matched cohorts would be required to deconvolute these factors”

We believe the limitations are made clear and that the conclusions are suitably within these limitations. We also believe that the comparisons still provide insight into the effects of heterogeneous antibody profiles on complement-enhanced neutralisation.

The description of the microneutralization assay should clarify the volume and final concentration of serum. Were different volumes and concentrations used for the different cohorts?

We have now included the volumes and final concentrations for the serum in the relevant methods sections. The volumes remained the same for each of the cohorts

but the serum concentrations varied depending on the cohort or the viral strain used. The OCTAVE cohort had significantly lower neutralisation and so higher starting concentrations of sera were required to neutralise the virus and calculate the 50% neutralisation titres.

The relevant sections on lines 223 – 229 now reads:

“Vaccinee serum samples were serially diluted 1:2 across 12 (OPTIC cohort starting dilution: 1:80 against VIC01 or 1:10 against BA.1) or 7 (OCTAVE cohort starting dilution: 1:10 against VIC01) dilution points, in a volume of 10 μ l per well. Then 10 μ l of HI-FCS, PHP, or equivalent volumes of assay media, were added to each dilution point in duplicate, for a final concentration of 20%. SARS-CoV-2 VIC01 (OPTIC and OCTAVE cohorts) and BA.1 (OPTIC cohort) strains were added to each well at a final concentration of \geq 100 FFU, for a final volume of 40 μ l.”

Why was a different amount of virus used for the assays with HI-PHP (modification 1)?

The readout of these assays depends on achieving an optimal number of foci i.e. between >50 to ~800 foci to normalise the results and plot the 4-parameter logistic curve. Because this assay was modified to remove the virus supernatant 2 hrs post-infection, the infection efficiency was less than the conventional MNA, and this reduced the total number of foci for the assay readout. Therefore, more virus was required at the start of the experiment to achieve the desired readout of foci for the analysis. However, the final target readouts of the foci number were similar.

We have adjusted the wording in the manuscript to more clearly address this point and to keep it consistent with the rest of the methods.

This section on lines 255 - 257 now reads:

“The SARS-CoV-2 VIC01 strain was added to each well at a final concentration of \geq 100 FFU and the samples were incubated for 1 hr at 37°C, 5% CO₂.”

The possibility of complement contributing to cell lysis was not explored, calling into question the conclusion that blocking of virus-host receptor interactions is largely responsible for enhanced neutralization.

We respectfully disagree, as we believe there are multiple points of evidence in the manuscript and supplementary files that demonstrate that the mechanism was not cell lysis. Firstly, supplementary figure 4 (previously supplementary figure 2) shows the complement-mediated cytotoxicity following incubation of the complement source with both Vero E6 cells and Calu-3 cells. There is no visible lysis of the cell monolayer in either condition, which replicates the conditions used for the microneutralisation assays up to a 24-hr incubation period. Thus, the complement source did not contribute to cell lysis. A possible counter-argument to this could be that whilst complement alone does not contribute to cell lysis, the complement source in the presence of the immune sera during SARS-CoV-2 infection can lead to

cell lysis i.e. ADCC antibodies binding the SARS-CoV-2 spike presented on infected cells could induce complement-mediated cytotoxicity. However, we address this possibility during the quality control process when analysing the MNAs. Each well on each plate is scanned using the BioSpot™ Software Suite of the ImmunoSpot® (Cellular Technology LTD) which allows visualisation of the cell monolayers to ensure integrity and to count foci. We did not observe damage to the cell monolayer in any of the conditions tested and every sample was subjected to the same QC process.

We have now added this information into the manuscript in the relevant methods section for MNAs on lines 297 – 299, which now reads:

“Foci were automatically counted using the BioSpot™ Software Suite and subjected to quality control to verify the counting accuracy and ensure integrity of the cell monolayer.”

For statistical analyses and calculation of standard errors, were duplicate samples included? In other words, if 6 replicates were done across two assays, did $N = 6$? 12 ? 2 ? Since the replicates would not be independent, it seems that N should equal 2 for the scenario described—unless repeated measures were factored in.

To use the same example, 6 replicates across two assays would be $n = 6$. All measures were then included in the final calculations, including all repeated measures. Therefore, $n = 6$.

Reviewer #3 (Remarks to the Author):

This study shows that as is the case for other viral diseases, replenishment of the complement system significantly increases SARS-CoV-2 antibody neutralisation titres of vaccinee serum. This study is important as it suggests that heat inactivation of plasma samples underestimates the neutralization capacity of antibodies against SARS-CoV-2. The observation that this varies with cell type viral strains, and the immune sera, also points to mechanism but this is not adequately addressed by this study. Theories are provided but in my opinion need to be experimentally explored in order to warrant publication.

Major points:

1. Experimental evidence should be provided to understand why different cells show different outcomes ie blocking of TMPRSS2, and why different cohorts exhibit different levels of complement mediated neutralization through depletion or measurement of IgG3 for example.

Thank you for your comments. We have now performed additional experiments to further explore the mechanism of complement-enhanced neutralisation. Firstly, we have conducted experiments comparing the effects of Vero E6 cells (no TMPRSS2)

with Vero E6 cells expressing TMPRSS2, using PHP and HI-PHP with all OPTIC serum samples at a fixed 1:1000 dilution. There was only enough remaining sera and complement to conduct this experiment once with each sample tested in triplicate. There was no significant difference in NT50 between the two cell lines, although some samples showed greater heterogeneity in responses between cell lines than others. It's possible that NT50s are affected in only some of the samples by TMPRSS2 but the impact isn't large enough to be detected with the current statistical power.

Line 781 – 785:

“Also, we did not observe a significant overall difference in NT50s of the OPTIC cohort using Vero E6 cells compared to Vero E6 cells constitutively expressing TMPRSS2 (**Supplementary Figure 8**). However, given the limited sample size due to serum limitations, it's possible that individual sample differences may be masked by the overall population.”

We have also now measured the levels of SARS-CoV-2 spike-specific IgG1, IgG2, IgG3, and IgG4 in all of OPTIC and OCTAVE serum samples used within this study. We have created a new results section to include this work and the relevant section in the results can be found below. These results were then correlated with ADCD (described later) and neutralisation:

Lines 683 – 703:

“IgG1 and IgG3 subclasses are reportedly the most potent activators of the complement system, followed by IgG2, then IgG4 with minimal activity reported. Firstly, total anti-SARS-CoV-2 spike IgG titres were significantly higher in the Enhanced cohort ($p = 0.003$) (**Figure 6 (a)**). Both cohorts showed the same trend with the highest measurements for IgG1 binding, followed by IgG2, then IgG3, then IgG4 (**Figure 6 (b)**), with significantly higher IgG1 ($p = 0.033$), IgG2 ($p = 0.009$), and IgG3 ($p = <0.0001$) titres in the Enhanced group (**Figure 6 (c)**). In spite of the differences in total IgG titres, there was no significant difference in IgG4 ($p = 0.188$) titres, which suggests that this subclass constitutes a higher proportion of total IgG in the Non-Enhanced group. Lastly, ADCD (measuring C3c deposition) in response to the SARS-CoV-2 spike protein was significantly higher in the Enhanced cohort ($p = <0.003$) (**Figure 6 (d)**). To understand the relationships between total IgG, IgG subclass, ADCD, and complement-enhanced neutralisation, all conditions were correlated using Pearson correlations (Benjamini Hochberg false discovery rate of 0.05) (**Figure 6 (e)**). The most significant correlations for the Non-Enhanced cohort were for total IgG, ADCD, and neutralisation, whereas the Enhanced cohort was total IgG and neutralisation only. The Non-Enhanced cohort also showed significant correlations of IgG4 titres with neutralisation, which was not observed in the Enhanced cohort. There were no clear correlations that could distinguish between HI-FCS and PHP supplemented neutralisation titres in either cohort to provide insight into complement-mediated enhancement.”

Described in more detail in response to comment 2, we also utilised multiple machine learning techniques to probe the dataset further and found that IgG3 was

identified in all models as being the most important feature for influencing complement-enhanced neutralisation from our observations.

We thank the reviewer for their suggestions and we believe we have sufficiently addressed their request for additional experiments. We believe these experiments have added value to the study by further exploring mechanism (TMPRSS2) and the antibody characteristics which influence this (IgG subclass and ADCD), that we were then able to probe further using machine learning. We also feel it is important to highlight that the differences observed between cell lines within this study is a strength, even if a single, precise mechanism is not defined. Many of the studies which show a complement-mediated enhancement of neutralisation do so in only one cell line, with the one exception by Li et al 2017 that we are aware of. We therefore believe that the use of 3 cell lines in this manuscript is a real asset.

2. It is not clear why the authors insist that complement enhanced neutralization is somehow separate to Fc effector function as this is the region that complement would bind to on antibodies and one of their proposed explanation for differences between the cohorts are all the mechanisms that define Fc function. Using plasma samples in this study the authors should perform ADCD and note if there are any correlations. This may just be a description of the ability of the antibodies to deposit complement.

We would firstly like to clarify that we do not insist that enhanced neutralisation is separate to Fc effector functions and this is why some of our proposed explanations relate to Fc function. We believe the confusion may come from our following statement on lines 80 and 81: "This phenomenon is independent of other immune factors involved in Fc-mediated functions"? Here, we try to clarify that complement-enhanced neutralisation in these examples is not dependent on cell-mediated functions such as opsonisation/phagocytosis, which are commonly attributed to the role of complement in neutralisation. In our experience, people are often unaware of the mechanisms through which complement can enhance neutralisation through direct interactions with antibodies and this is what we're trying to highlight.

To try and avoid this confusion in future, we have now changed this section which can now be found on lines 80 - 82:

"This phenomenon is independent of other immune factors such as opsonisation and phagocytosis²⁷."

We have now also conducted ADCD assays for all of the OPTIC and OCTAVE samples used within this study, with the exception of one OPTIC sample where there wasn't enough serum left. Please note we also had to use a new IgG/IgM depleted complement source to complete this set of experiments as we exhausted the stocks of original complement source that was used in this study. We found that ADCD was significantly higher in the samples which showed complement-enhanced neutralisation. For the Enhanced group only, ADCD did not significantly correlate with neutralisation but for the Non-Enhanced group, ADCD, total IgG, and

neutralisation all significantly correlated (Figure 6). The relevant results discussion is on lines 683 – 703 as already copied in response to comment 1.

Using previously collected data for antibody epitope binding and total IgG, combined with the additional requested experiments for IgG subclasses and ADCD, we utilised various machine learning models to further understand which of these factors are most important for determining complement-mediated enhancement of neutralisation. Using a random forest analysis, IgG3 was determined to be the most important factor for predicting outcome with a positive relationship to enhancement. The model performed with an accuracy of 77.5% which was consistent across 20 iterations of the model (14.5% CV). The area under the curve measuring sensitivity and specificity was 0.864 with 8.5% CV across the 20 iterations which also suggests the model performed well. This was supported by multiple logistic regression with LASSO and ridge methods which both highlighted IgG3 as the most important predictor. The relevant results section is copied below:

Lines 713 – 731:

“To understand which of these antibody characteristics might be important for the complement-mediated enhancement of neutralisation to occur, and if a combination of factors are required, we performed a supervised random forest (RF) machine learning algorithm and LASSO and ridge regression. The RF model classified whether a sample showed a significant complement-mediated enhancement of neutralisation with a mean accuracy of 77.5% and a 14.5% CV across 20 iterations. The model’s ability to separate positive and negative cases across all classification thresholds as measured by the area under the curve (AUC) was 0.864 with 8.5% CV, suggesting it’s fit for purpose (**Supplementary Figure 7**). The most important feature for model accuracy (**Supplementary Figure 7**) and node purity was IgG3 (**Figure 6 (f)**). We then performed LASSO and ridge regression analyses using the same measurements of antibody characteristics as verification with a separate model. SARS-CoV-2 spike specific IgG3 titres were again highlighted as the most important factor in both models and was positively associated with complement-enhanced neutralisation (**Figure 6 (g)**). LASSO regression highlighted four important variables (SARS-CoV-2 spike specific IgG3 [COEFF 2.76], HKU1-spike specific IgG [COEFF 1.00], SARS-CoV-2 spike specific IgG1 [COEFF 0.43], SARS-CoV-1 spike specific IgG [COEFF -0.17]) and ridge regression highlighted two (SARS-CoV-2 spike specific IgG3 [COEFF 1.18] and HKU-1 spike specific IgG [COEFF 0.52]).”

We then discuss these results further in the discussion section:

Lines 831 – 844:

“We found significant increases in total SARS-CoV-2 spike-specific IgG, IgG1, IgG2, IgG3, ADCD, and differences in Coronavirus protein binding characteristics for samples that showed a complement-mediated enhancement of neutralisation. IgG4 has a mostly inert role in complement utilisation²⁶ and this constituted a higher proportion of total IgG in the Non-Enhanced group compared to the Enhanced group. Machine learning approaches with RF and both LASSO and ridge regression models highlighted IgG3 as the most predictor from the observations tested, with a positive relationship to complement-enhanced neutralisation. We believe this further supports

our findings and hypothesis that the mechanism is primarily mediated through the binding of proteins C1-C3, as IgG3 has the strongest affinity for C1q binding out of the IgG subclasses⁵⁶. Future studies could expand on this using purified antibodies with the same epitope recognition, that differ in the IgG subclass. Similarly, to experimentally address the significance of epitopes, antibodies with the same IgG subclass that differ in epitope recognition could be used.”

Minor queries

1. Worth looking at <https://www.nature.com/articles/s41591-024-03131-2> for the correlates of protection description

Thank you for the suggestion. We have now amended the first introductory paragraph to better align with the description of correlates of protection given within the suggested reference. This now reads:

Lines 63 – 72:

“The implementation of COVID-19 vaccines has proven highly effective against the development of severe disease, hospitalisation, and death. There is a good correlation between antibody binding and antibody neutralisation with protection against disease, but this can change within the context of viral evolution and emerging variants¹, with further complexity in correlating the impact of Fc effector functions². With concerns regarding breakthrough infections, a lack of therapeutics, and ongoing attempts to develop novel vaccines to combat the continued emergence of new variants, clearly defined and ongoing assessments of correlates of protection are imperative.”

2. I find the extensive explanation of all the complement pathways unnecessary but feel the 3 mechanisms of complement enhanced neutralization underexplained.

We have now expanded on the explanation of the four proposed mechanisms which we hope provides further clarity. The following statement was added to the introduction on lines 115 – 131:

“The aggregation of virus particles by antibody binding can cause a reduction in viral attachment to host cells. The formation of viral aggregates can be enhanced by the deposition of complement proteins following antibody binding, which usually depends on proteins C1-C3^{5,8,24}. The second mechanism, the inhibition of viral attachment/entry to host cells, refers to the masking of the viral antigens required for infection through the deposition of complement proteins. In addition to antibody binding, the subsequent binding of C1, C4, C2, and multiple C3 molecules (where up to 1000 C3 molecules may be cleaved by one C3 convertase) increases the chances of blocking protein-protein interactions required for infection^{2,4,28,29} or reducing the stoichiometric threshold for antibody-mediated neutralisation³⁰. The third mechanism of viral lysis requires the complete activation of the complement system, resulting in the formation of the MAC. The MAC may lyse the lipid membranes of enveloped viruses, thus reducing their infectivity^{31,32}. The fourth mechanism also depends on complete activation of the complement system. Antibodies can bind to viral antigens expressed on the surface of infected cells, leading to complement deposition and formation of the MAC to lyse the infected host cells and reduce viral titres^{21,24}.”

3. Can the authors show the ELISA validation of the “healthy” complement showing

that these individuals were not infected with SARS-CoV-2? Was nucleocapsid also run?

This data is already published in the reference provided within the manuscript (Mellors, J., 2022. Understanding the Role of the Complement System in Ebola virus and SARS-CoV-2 pathogenesis). The samples were tested for reactivity to SARS-CoV-1 and SARS-CoV-2 spike proteins. We did not test for nucleocapsid as antibody binding to the spike protein was deemed the most relevant interaction for these studies of neutralisation, although we understand that screening for nucleocapsid could help verify absence of previous infection. As we did not observe any effects on SARS-CoV-2 neutralisation with the complement source alone, taken together with the ELISA validation, the reagent was deemed suitable for use.

4. How were the dilutions for the PHP decided?

The dilutions were chosen based on a combination of pilot experiments, the use of this complement source in previous studies, and what's commonly used in the literature. We found 20% was the optimal dilution to use as it did not cause cytotoxicity, it enabled us to observe differences in our sample cohorts, it remained a physiologically relevant concentration, and it would provide enough sample volume to complete this study.

5. As neutralization is highly specific to samples, data presented In figure 2 should be shown in a table or heatmap where all samples are individually indicated or fold changes shown in the plot.

This information is already available in supplementary Table 3 which includes the NT50 for all of the individual samples in all of the conditions tested, along with 95% confidence intervals, the log2 fold-change between conditions, and whether the difference was significant via the sum-of-squares F test. We have also already done the same for all of the OCTAVE samples and this information is available in Supplementary Table 4. References to these files within the manuscript are as follows:

Lines 481 – 482:

“All NT50 values for the OPTIC cohort can be found in **Supplementary Table 3.**”

Lines 642 – 643:

“All NT50 values for the OCTAVE cohort can be found in **Supplementary Table 4.**”

We have now also included a 'Source Data' file containing all of the raw data used to generate all figures within the manuscript and supplementary files.

6. Can the authors show neutralization assay in presence of PHP alone? IE no antibody/sample

As the PHP alone doesn't neutralise virus in the absence of antibody/serum, this cannot be shown in a neutralisation assay. However, Supplementary Figure 3 (formerly Supplementary Figure 1) should provide the information required which compares the raw foci number of the control conditions tested (media, HI-FCS and PHP in absence of serum) across the entire study that were then used for normalisation on each plate. This shows that the addition of HI-FCS or PHP did not reduce the number of foci (viral infection) compared to the media-only control, thus showing no neutralising activity. Reference to this can be found in the main text.

Lines 458 – 460:

“Also, the use of HI-FCS or PHP did not demonstrate virus neutralising activity in the absence of OPTIC vaccinee serum (**Supplementary Figure 3**)...”

7. Does the 4500 dilution and the 1500 dilution correlate in Figure 3? Because several sample look discordant in their differences.

To confirm this in response to your question, we ran a pearson correlation of the 1:1500 and the 1:4500 dilution in the PHP condition which shows a significant correlation ($p = 0.0043$) with an r value of 0.8119. We were unable to do the same for the HI-PHP condition as some of these samples are below the limit of detection in the 1:4500 condition. So yes, the two dilutions correlate.

8. The authors show that the TEM did not significantly differ, however it is unclear how they make that assessment. Are multiple views somehow quantified in an unbiased way?

Thank you for your comment. We have now changed the wording of this as we did not conduct a statistical analysis to determine this and have replaced the term “significantly” with “noticeably”. This now reads:

Line 567 - 570:

“Negative staining of the samples revealed that the addition of PHP did not noticeably differ in the lysis of virus particles or the formation of viral aggregates from the use of HI-PHP, or the virus-only control, using OPTIC serum samples 8 (**Figure 4 (a)**) and 10 (**Supplementary Figure 5**).”

We have also changed the wording in the discussion on line 812-813 to read:

“Our observations via TEM showed an absence of any obvious viral aggregation and lysis...”

Our conclusions were independently made and verified by two of the authors with a total of 136 images captured across the two OPTIC serum samples at 3 different magnifications. As this experiment was conducted before any indication of mechanism was known, there was no prior expectation as to what the results would be i.e. lysis, aggregation, or no response. Examples where virus aggregation has been previously reported (Johnson et al., 2008), the effect is clear and, in some instances, requires a 10x reduction in magnification to observe the formation of such

large aggregates. We did not observe anything like this. Observations of viral lysis are less prominent and so we only report that clear differences were not observed. These interpretations were later supported by our C3 inhibition studies which showed that significant enhancement could still occur in absence of C3 and downstream proteins, albeit to a reduced capacity.

9. It seems that the mechanism of influence of complement is not clear cut, with only a limited decrease seen in ACE2 binding. Are there alternate mechanisms previously undiscovered that may explain this?

We agree that only a limited decrease is observed with ACE2 binding and that this alone does not fully explain the mechanism, as described within the manuscript. If we understand the question correctly, the reviewer is asking whether mechanisms other than those described within this manuscript may be responsible? In which case yes, it's entirely plausible that an alternate mechanism is also influencing these findings. Our explanations are based on extensive reading of the literature and what's commonly reported. But as there's relative paucity of studies describing mechanism – particularly for Coronaviruses – it's possible that other mechanisms exist. However, this would likely still be confined by what we know already about the complement system, which is more extensive.

To address the possibility that alternate mechanisms may be responsible beyond what we've currently identified, we have included the following:

Lines 897 – 900:

“Given the large heterogeneity between the samples tested in this study, it's possible that multiple or alternate mechanisms may be responsible for the complement-enhanced neutralisation.”

REVIEWER COMMENTS

Reviewer #2 (Remarks to the Author):

How sensitive is the BioSpot software for determining the effects of antibody and complement on cells?

Thank you for your question. The effects of antibody and complement on cells would be apparent through cell lysis and visible gaps in the cell monolayer, similar to a plaque assay. As a general reference, we typically set the minimum gating for foci counting in our SARS-CoV-2 MNAs as 0.0016mm^2 . These small gates can be visualised by eye within each entire well during the QC process. A typical eukaryotic cell is reportedly $25\ \mu\text{m}$ in diameter. So, the effect of antibody and complement on just a few cells should be detectable. Also, if cell lysis occurred in response to the viral infection, we would expect lysis to occur at the site of the foci and therefore it would not be possible to observe and quantify them. This is not to say that ADCC does not occur in response to SARS-CoV-2 infections, just that it was not observed in our experimental conditions.